# The genomic landscape of Mexican Indigenous populations brings insights into the peopling of the Americas

Humberto García-Ortiz [1,15 ✉], Francisco Barajas-Olmos[1,15], Cecilia Contreras-Cubas[1,15], Miguel Ángel Cid-Soto[1], Emilio J. Córdova[1], Federico Centeno-Cruz [1], Elvia Mendoza-Caamal[1], Isabel Cicerón-Arellano[1], Marlen Flores-Huacuja[1], Paulina Baca[1], Deborah A. Bolnick [2], Meradeth Snow[3], Silvia Esperanza Flores-Martínez[4], Rocio Ortiz-Lopez [5], Austin W. Reynolds[6], Antonio Blanchet[1], Mirna Morales-Marín[1], Rafael Velázquez-Cruz[1], Aleksandar David Kostic [7], Carlos Galaviz-Hernández [8], Alejandra Guadalupe García-Zapién[9], José Concepción Jiménez-López[10,16], Guadalupe León-Reyes[1], Eva Gabriela Salas-Bautista[10], Blanca Patricia Lazalde-Ramos[11], Juan Luis Jiménez-Ruíz [1], Guadalupe Salas-Martínez[1], Jazmín Ramos-Madrigal [12], Elaheh Mirzaeicheshmeh[1], Yolanda Saldaña-Alvarez[1], María del Carmen Abrahantes-Pérez [1], Francisco Loeza-Becerra[13], Raúl Mojica-Espinosa [1], Federico Sánchez-Quinto[1], Héctor Rangel-Villalobos[14], Martha Sosa-Macías[8], José Sánchez-Corona[4,16], Augusto Rojas-Martinez [5], Angélica Martínez-Hernández[1] & Lorena Orozco [1 ✉]

The genetic makeup of Indigenous populations inhabiting Mexico has been strongly influenced by geography and demographic history. Here, we perform a genome-wide analysis of 716 newly genotyped individuals from 60 of the 68 recognized ethnic groups in Mexico. We show that the genetic structure of these populations is strongly influenced by geography, and our demographic reconstructions suggest a decline in the population size of all tested populations in the last 15–30 generations. We find evidence that Aridoamerican and Mesoamerican populations diverged roughly 4–9.9 ka, around the time when sedentary farming started in Mesoamerica. Comparisons with ancient genomes indicate that the Upward Sun River 1 (USR1) individual is an outgroup to Mexican/South American Indigenous populations, whereas Anzick-1 was more closely related to Mesoamerican/South American populations than to those from Aridoamerica, showing an even more complex history of divergence than recognized so far.

A full list of author affiliations appears at the end of the paper.

Mexico has long acted as a natural bridge for human migration from North America to Central and South America and vice versa. Together with historical events, these movements have been crucial in shaping the genetic makeup and structure of populations in the Americas[1–5]. There has been great interest in understanding the genetic structure of Native American populations, partly because studying these populations has been helpful in elucidating aspects of the global dispersal of modern humans[2,4–6]. Today, the 68 recognized ethnic groups in Mexico are clustered into 11 linguistic families[7], with unique customs and cultures. These populations can be divided into two main geographic/cultural areas: Mesoamerica and Aridoamerica. Mesoamerica comprised central and southern Mexico, and during the pre-Hispanic era was inhabited by sedentary agricultural societies favored by the great biodiversity of this region. Aridoamerica encompassed a semiarid area in northern Mexico that preserved nomadic forms of subsistence throughout the pre-Hispanic era.

Despite being one of the largest and most diverse groups in America, the Mexican Indigenous populations are still underrepresented in terms of the number of genotyped individuals and geographic regions sampled. To the best of our knowledge, few Mexican ethnic groups have been examined at the genome-wide level, yet a complex genetic structure has been observed in such groups[2,5,8]. Therefore, key questions about the genetics of Mexican Indigenous populations remain unsolved.

In this work, we perform a population genetics study by genotyping at the genome-wide level 716 individuals from 60 of the 68 recognized ethnic groups in Mexico belonging to the Metabolic Analysis in an Indigenous Sample (MAIS) cohort[9–11], which were merged with previously published data sets, yielding a total of 1086 Native Americans from Mexico, representing all linguistic families except Kickapoo (Algonquian language family) (Fig. 1a and Supplementary Table 1). We find that the genetic structure of Mexican indigenous populations is influenced by geography and geographic barriers, historical events, such as the establishment of sedentary agriculture in Mesoamerica, or European contact. Finally, comparisons with ancient genomes from America show that populations from Aridoamerica and some from Mesoamerica may carry an additional ancestry from an unknown population related to the SNA/ANC-A branch that split above the Anzik-1 individual.

## Results

### Genetic variation and population substructure in Mexican Indigenous groups is influenced by geography

First, we compared our 716 Mexican Indigenous individuals from 60 ethnic groups (72 communities) with 146 previously published populations worldwide[2,8,12], including Mexican Native American populations previously reported by Reich et al.[8], Moreno-Estrada et al.[2], and Silva-Zolezzi et al.[12]. The merged data set comprised 3490 individuals from 218 populations and 61,393 autosomal SNVs. Principal component analysis (PCA)[13] indicated that Mexican Indigenous populations clustered with other Native American groups from North and South America (Supplementary Fig. 1). On the other hand, admixture[14] analyses assuming $K = 4$ clusters showed that some Native American individuals are admixed with European and African populations, which is consistent with the history of the Mexican populations. We detected 325 Indigenous samples from the MAIS cohort with at least 0.99 Native American ancestry (Supplementary Fig. 2, upper panel).

In order to minimize the effects of recent admixture on our simulations, we performed local ancestry inference using RFMIX[15] in each data set, except for Reich et al.[8] data set as detailed in the "Methods" section. Non-Native ancestry tracks were masked in the individuals from Indigenous populations, and the masking accuracy was assessed by running the admixture analyses again assuming $K = 4$ clusters (Supplementary Fig. 2, lower panel).

Next, to assess the genetic structure of the Mexican Indigenous populations without the recent European and African ancestry, we combined the masked genomes of the Mexican Indigenous individuals from the MAIS cohort with the data sets from Reich et al.[8], Moreno-Estrada et al.[2], and Silva-Zolezzi et al.[12], yielding a total of 1086 individuals. PCA in the whole Mexican Indigenous masked data set showed that the first axis of variation discriminated the Indigenous Mexican populations from the North, mainly groups from Aridoamerica, from those of Mesoamerica in the Center/South and Southeast (Fig. 1b). We also found a correlation between PC1 and the longitude and latitude (Supplementary Fig. 3a, b), and a Mantel test showed a significant correlation between genetic and geographical distances ($p = 0.001$, $r = 0.63$, Supplementary Fig. 3c). Moreover, PCA of Mesoamerican populations showed that the first two axes of variation separated the populations from the Center/South and Southeast following a geographic pattern (Fig. 1c). These results suggest that geographic location influences the genetic structure of these populations.

Furthermore, pairwise-$F_{ST}$ comparisons identified the Tarahumara, Pima, Guarijio, and Cucapa in northern Mexico, and previously published populations, such as the Seri (North) and Lacandon (Southeast)[2], had the highest levels of genetic differentiation when compared with the other populations based on this statistic (Fig. 2a and Supplementary Data 1). These observations suggest that these populations have experienced higher degrees of isolation or genetic drift, and possibly various founder effects that amplified this drift.

A midpoint rooted neighbor-joining (NJ) tree based on the pairwise-$F_{ST}$ population distances showed a correlation between genetic structure and geographic distance, independently of the linguistic classification (Fig. 2b). The NJ tree topology revealed five major regions, with high clustering of the ethnic groups according to their geographic location. Furthermore, several ethnic groups from different regions are genetically closer to their geographical neighbors even if they belong to different linguistic families. For example, the Nahuatl from San Luis Potosi (Yutonahua), Pames (Oto-mangue), and Huasteco (Mayan) coinhabiting the Huasteca region fall into the same clade from the NJ tree. Similarly, the Mixe (Mixe-zoque) inhabiting Oaxaca are closer to Oto-mangue linguistic family groups from Oaxaca and the Zoque (Mixe-zoque) from Chiapas are closer to Mayan linguistic family groups (Fig. 2a, b).

To better understand the genetic composition of Mexican Indigenous populations, we carried out a genetic clustering analysis with the unsupervised model algorithm ADMIXTURE[14] using $K = 2–16$ clusters (Supplementary Fig. 4). The cross-validation procedure showed that, within the Mexican Indigenous populations, the $K = 9$ yields the lowest cross-validation error (Supplementary Fig. 5). Based on this $K$, six of these clusters were mainly observed in a single population (Seri, Tarahumara, Pima, Tepehuano, Huichol, and Lacandon). On the other hand, two of the clusters were mainly observed in several ethnic groups inhabiting the Center and South (here referred to as multiethnics), principally in populations from the Oto-mangue linguistic family, and the other cluster in populations from the Southeast that are part of the Mayan linguistic family. We observed that the multi-ethnic and Mayan components had opposite gradients, where the Mayan component was the most prevalent in the Southeast and the multi-ethnic components were more prevalent in the Center and South of Mexico (Fig. 2c and Supplementary Fig. 6).

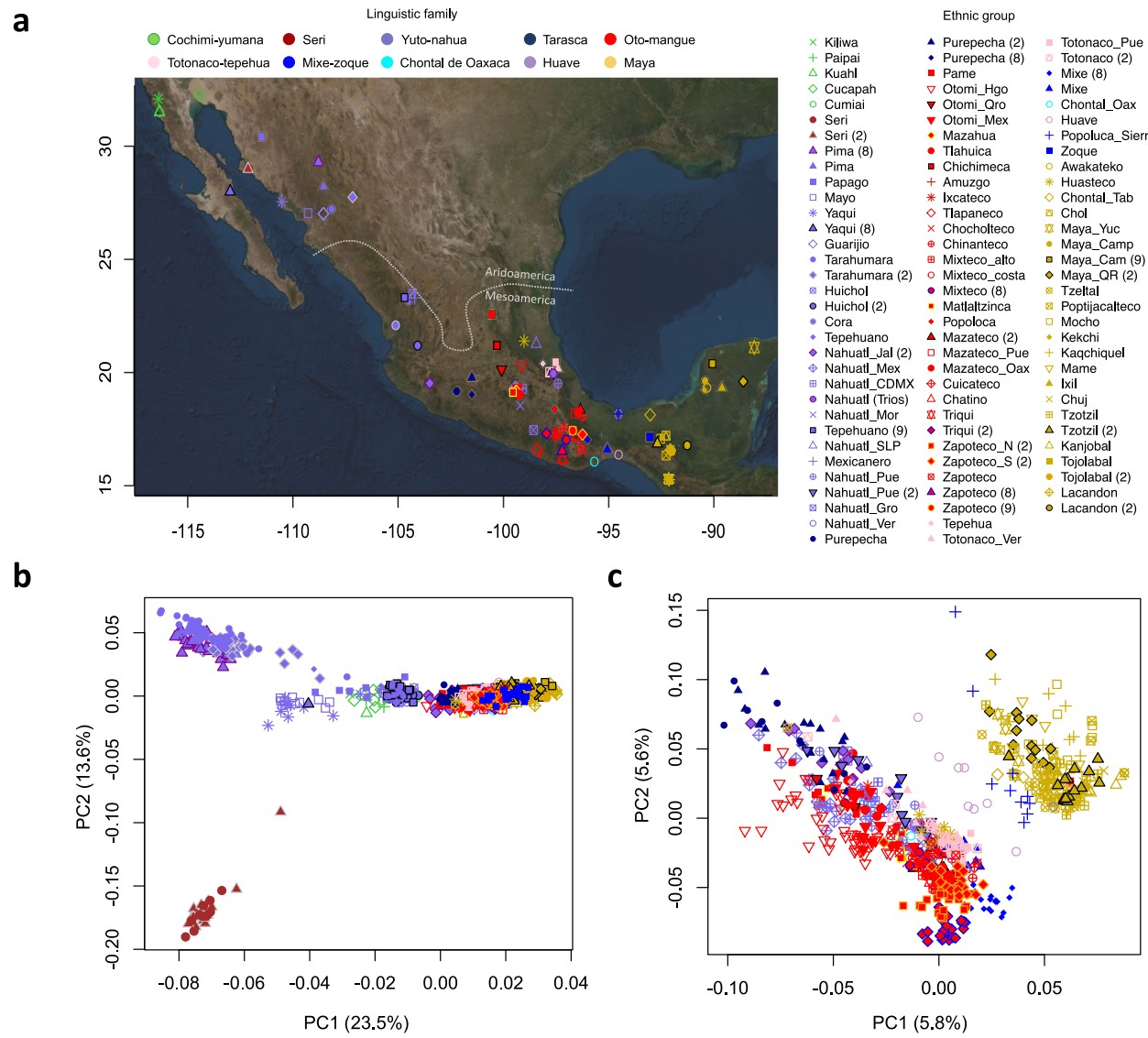

**Fig. 1 Geographic distribution and genetic relationships of Indigenous populations in Mexico. a** Map with locations of the sampled communities. The points in the map denote the approximate location where each ethnic group samples were collected. **b** PCA projection of all Mexican Indigenous groups tested, with the first two axes of variation differentiating ethnic groups from Aridoamerica and Mesoamerica. **c** PCA projection of Indigenous groups from Mesoamerica that resembles the geography of central and southern Mexico. Dot shapes denote the ethnic group and color the linguistic family according to INALI classification. Numbers between brackets are the corresponding references.

**Effective population size and divergence time estimation**. To track the demographic histories of Indigenous Mexican populations, we estimated the effective population size ($N_e$) across time based on two different methods. We included 48 ethnic groups from the masked data set, all of them with sample sizes of at least 10 individuals (Supplementary Table 2). Demographic reconstructions based on linkage disequilibrium (LD) analysis[16,17] showed little evidence of a fluctuation in $N_e$ before 150 generations ago (Supplementary Fig. 7). To evaluate more recent demographic changes, we estimated the $N_e$ based on identity by descent (IBD) tracks implemented in the IBDNe software[18,19]. We observed a decline in the $N_e$ between 15 and 30 generations ago in all tested populations that overlaps with the beginning of the European colonization of the Americas, followed by an expansion (Fig. 3a and Supplementary Fig. 8).

Next, we estimated the long-term $N_e$ based on LD patterns using Neon Software[16,17]. The long-term $N_e$ calculated in the whole sample set was 3169 (confidence interval of 2952–3402), which is similar to previous findings[5,20,21]. However, here we

documented a variation in the long-term $N_e$ between ethnic groups (Fig. 3b and Supplementary Table 3). The long-term $N_e$ was smaller in highly differentiated populations, such as Seris and Lacandons (984 and 1593, respectively). Other ethnic groups had a long-term $N_e$ between 1825 and 3331 individuals (Supplementary Table 3) and are similar to those previously reported in populations such like Tarahumara, Huichol, Triqui, and Maya[22]. The smaller long-term $N_e$ may have contributed to greater genetic drift and lower genetic diversity in these ethnic groups. To confirm this hypothesis, we inferred autozygosity using runs of homozygosity (ROH). As expected, the Seri and Lacandon groups had the highest proportion of the genome in ROH compared to the other populations tested, suggesting that the high genetic differentiation observed in these populations is due to genetic drift as previously reported[2] (Supplementary Fig. 9). We did not observe this phenomenon for other divergent populations, such as the Cucapa, Tarahumara, Guarijio, Tepehuano, and Huichol. In addition, the categorization of ROH by size showed that all tested Native American populations have a high proportion of

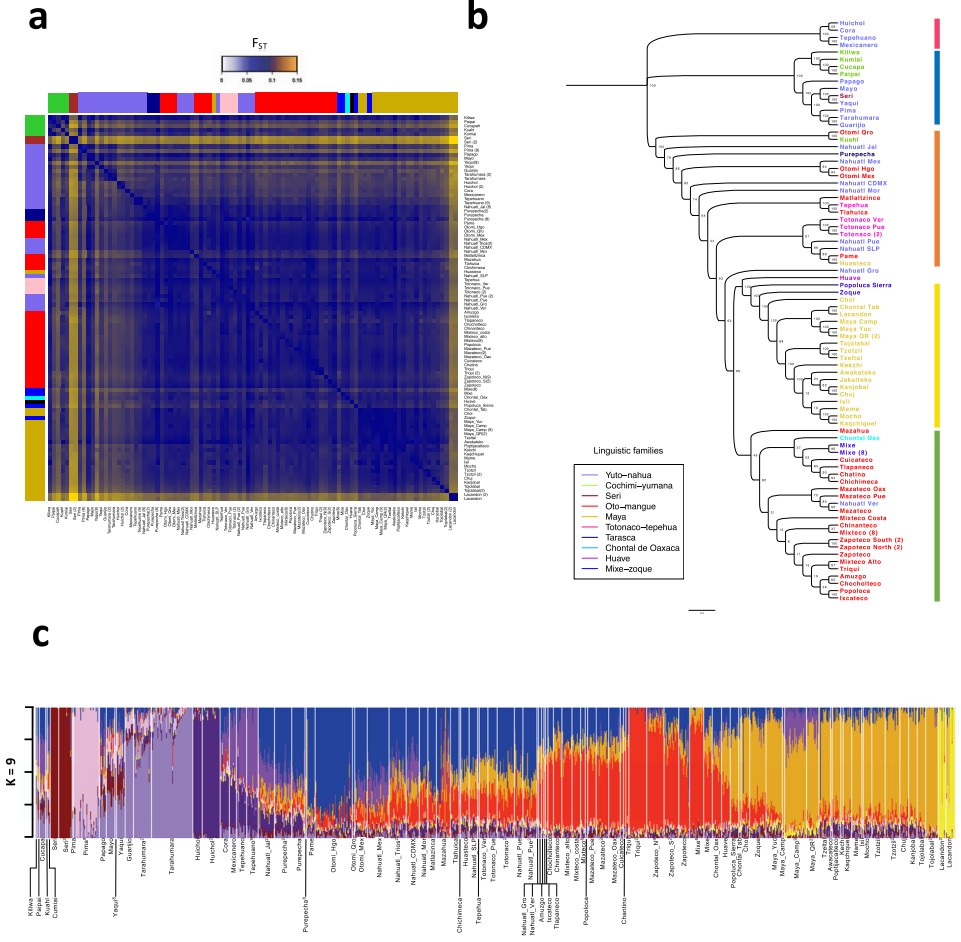

**Fig. 2 The genetic diversity of Indigenous groups correlates with their geographic distribution. a** Pairwise-$F_{ST}$ matrix for all tested populations. Colored bars represent the linguistic family. **b** $F_{ST}$-based neighbor-joining tree showing the correlation with geographic location independent of linguistic classification (colored names), the numbers above the branches indicate the bootstrap values. Colored vertical lines represent the identified geographic regions: North (blue), Northwest (red), Central-east (orange), South (green), and Southeast (yellow). **c** Admixture analysis assuming $K = 9$ clusters in Mexican Indigenous populations. Superscript numbers are the corresponding references.

short ROH (1–2 Mb), which is consistent with the fact that these populations have experienced a series of bottlenecks throughout their history[23,24]. Moreover, with the exception of Yaqui, Mazateco from Oaxaca, Chontal from Oaxaca, and Maya from Yucatan and Quintana Roo, we observed that all tested populations exhibited different proportions of ROH longer than 8 Mb (Supplementary Fig. 10), which is consistent with the presence of episodes of isolation and/or inbreeding[23,24].

Both the long-term $N_e$ and $F_{ST}$ between pairs of populations were employed to calculate the divergence time between populations in generations ($T$) assuming a clean split between them. To scale $T$ in years, we assumed 28 years per generation[25]. Seri and Lacandon populations have the highest $T$ values compared to other populations (Fig. 3c and Supplementary Data 2), and the uppermost value of $T$ was observed between Seri and Maya from Quintana Roo ($T = 11.8$ ka ago, Supplementary Data 2). Considering the ecogeographic region, we observed a higher $T$ between populations from different regions than those from the same region. Populations from northern Mexico corresponding to Aridoamerica diverged from the populations in the Center/South around 3.96–9.47 ka ago and from the Southeast populations ~4.84 to 10.15 ka ago (Fig. 3c, d and Supplementary Data 2).

To better understand the demographic connections among the Mexican indigenous populations, we performed an IBD analysis

in 325 individuals from our data set with >99% Native American ancestry using Hap-IBD[26] (see "Methods" section) (Supplementary Table 4). We also explored the ethnic group genetic affinities within and between different geographical regions according to those observed in the NJ tree and defined previously by Contreras-Cubas et al.[9]. In line with that observed with allele frequency-based methods (Supplementary Fig. 3), the IBD analysis also showed that the indigenous populations are related to each other following an isolation by distance model, both at the intra and interregional level. Therefore, in most cases, neighboring indigenous populations are more likely to relate to each other than to distant groups (Fig. 4 and Supplementary Fig. 11). At the intraregional level, this trend is exemplified by Tarahumara and Guarijio from North (Fig. 4a) or Chuj and Kanjobal from Southeast (Fig. 4e). Additionally, the shared IBD segment analysis revealed gene flow between Indigenous populations from different regions in Mexico. Some examples with shared IBD blocks were observed between Cora from Northwest and Zapoteco from South or Guarijio, Tarahumara and Seri from the North and Mayan groups from the Southeast (Supplementary Data 3–7).

An IBD analysis incorporating all populations per region using both intermediate (5–10 cM) or large (>10 cM) shared IBD blocks revealed possible spatiotemporal interaction dynamic patterns among indigenous groups. Intermediate IBD block sizes are

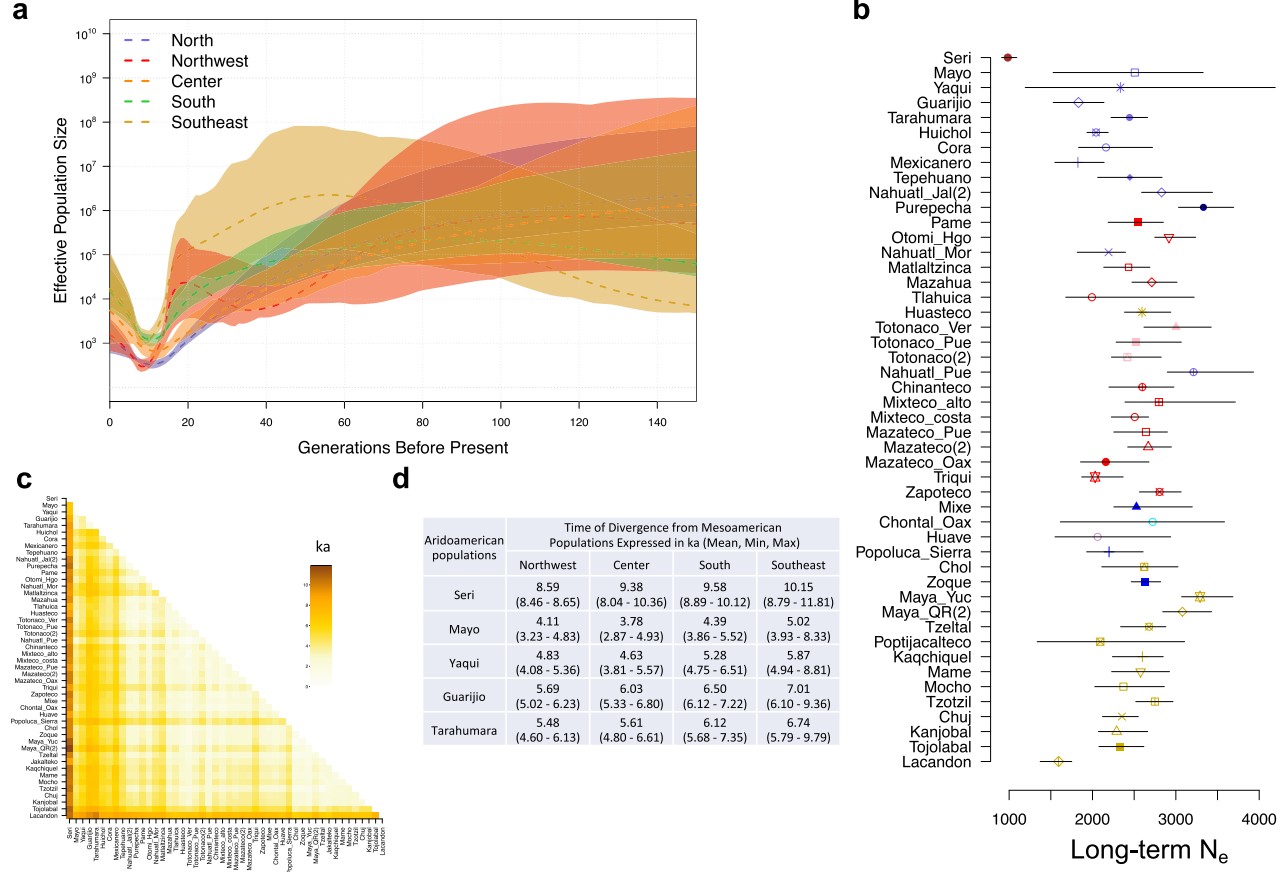

**Fig. 3 Effective population size and divergence time estimates. a** Demographic measure of effective population size across time showing a decline in population sizes in the five main geographic regions identified here: North, Northwest, Center, South and Southeast. **b** Long-term $N_e$ of all tested populations, shapes represent the long-term $N_e$ and errors bars represent the 95% confidence interval. Colors are according to the legends in Fig. 1. Numbers between brackets are the corresponding references. **c** Divergence time between pairs of populations. **d** Mean of the observed $T$ between Aridoamerican and Mesoamerican populations expressed in ka.

suggested to be dated to 500–1500 years ago (oldest), while large tracks are thought to be originated 0–500 years ago (youngest)[27].

Analysis of intermediate block sizes revealed that the Central East, South, and Southeast regions have older connections among them than do the northern regions (Supplementary Fig. 11a). Meanwhile, large IBD track analysis suggested that the North region has a more recent gene flow with Northwestern and Central East regions than do South and Southeast regions (Supplementary Fig. 11b).

**Genetic affinities between modern Native American populations and ancient inhabitants of the Americas.** To gain more insight into the early migration patterns, we compared the previously published genomes of the Anzick-1[28] and Upward Sun River 1 (USR1)[29] individuals with our data from the most representative sample of Indigenous peoples in the Mesoamerican and Aridoamerican regions of Mexico to date. First, we compared the ancient genomes with 59 worldwide populations and 325 individuals from our data set with at least 99% Native American ancestry (Supplementary Table 4) using an outgroup f3-statistic in the form of $f3$(Yoruba; Ancient, Modern). This analysis showed a high affinity of both ancient genomes with present-day Mexican Indigenous samples (Supplementary Fig. 12 and Supplementary Data 8).

We then combined the 325 indigenous samples with seven ancient genomes from American and South American populations[3,30] (Supplementary Table 5), yielding a total of

111,586 autosomal SNVs. A TreeMix tree on this data set placed the USR1 genome at the basal position of all Native American populations tested, including Anzick-1. Meanwhile, all Aridoamerican populations formed a separate clade from those formed by the Mesoamerican populations and Anzick-1 and the NNA/ANC-B branch. Similarly, a PCA including the ancient samples showed that the Anzick-1 genome is more closely related to Mesoamerican populations than Aridoamerican populations, whereas USR1 is placed as an outlier in the PCA space (Supplementary Fig. 13).

The TreeMix tree analysis suggested a deep divergence between populations in Aridoamerica and Mesoamerica (Fig. 5a and Supplementary Fig. 14) prior to the divergence between Mesoamerica and the Anzick-1 individual. This observation is inconsistent with $T < 10$ ka, as calculated based on $F_{ST}$ and long-term $N_e$ (Fig. 3c), and the fact that a TreeMix tree allowing 20 migration edges (Supplementary Fig. 14) and the IBD networks analyses (Fig. 4) exhibited multiple gene flow between Mesoamerican and Aridoamerican populations. To test this, we calculated a $D$-statistic in the form of $D$(Yoruba, NNA/ANC-B; AA, MA) using the Ancient Southern Ontario population Canada_Lucier[31] and Athabaskan[32] ancient genomes as representatives of NNA/ANC-B ancestry. We found these results to be consistent with $D \sim 0$, suggesting that the NNA/ANC-B populations are an outgroup for those from Aridoamerica and Mesoamerica from Mexico (Supplementary Figs. 15 and 16), which is consistent with the TreeMix tree. To test whether

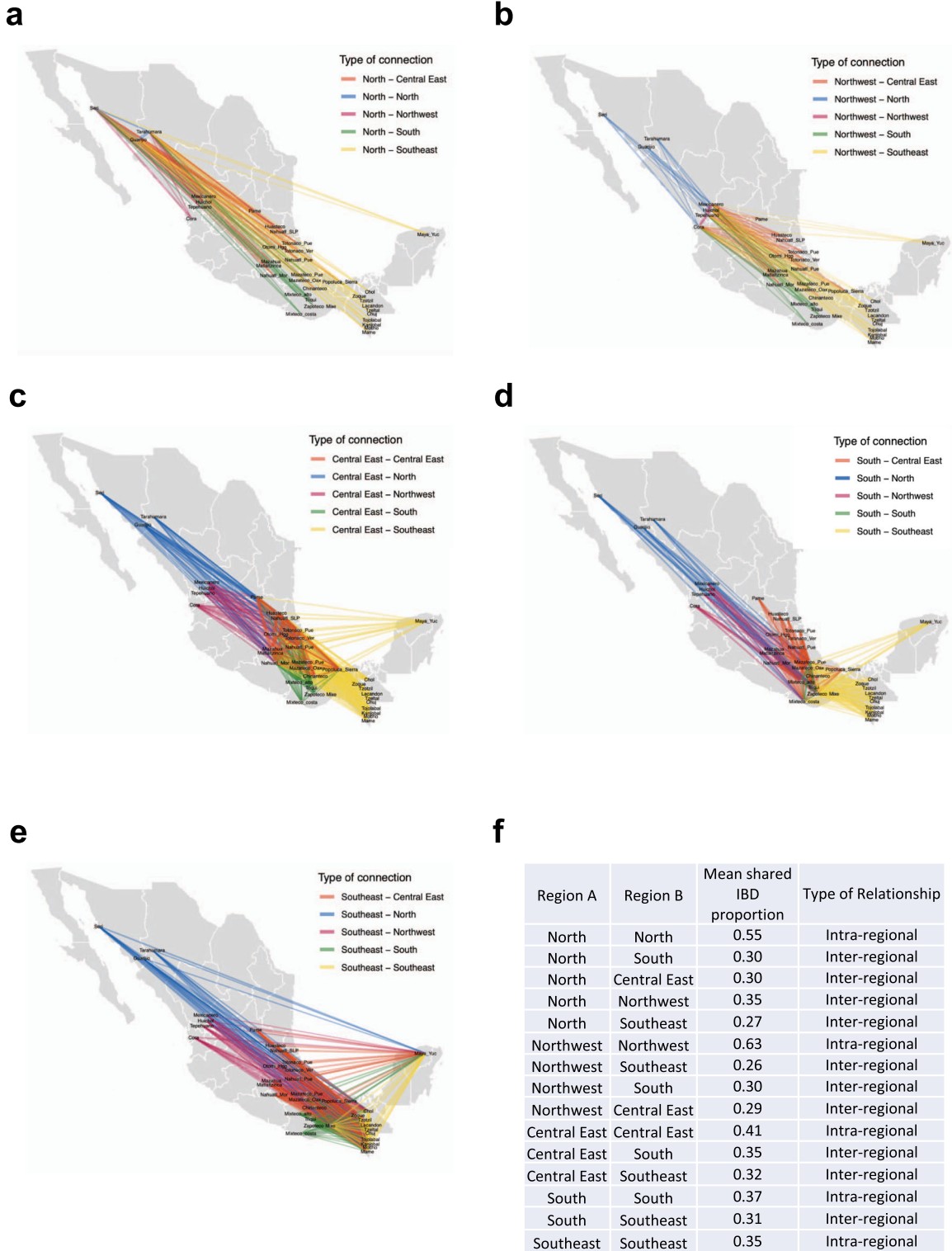

**Fig. 4 IBD segments analysis.** IBD segments analysis performed in Indigenous populations with at least 99% of Native American ancestry inferred from Admixture $K = 4$. Analyses were restricted for segments >7 cM. Shared IBD fragments shown proximal and distal connections between populations from the same and different regions. **a** North IBD segments, **b** Northeast IBD segments, **c** Central East IBD segments and **d** South IBD segments and **e** Southeast IBD segments. The width of each edge is proportional to the mean IBD length. **f** Table showing the mean values of IBD sharing within and between regions.

Mesoamerican populations form a clade with Anzick-1 to the exclusion of Aridoamerican populations, we estimated a *D*-statistic in the form of *D*(Yoruba, AA; Anzick-1, MA). We found that Mesoamerican populations share more alleles with

Aridoamerica than Anzick-1 shares with Aridoamerica (Supplementary Fig. 17).

To further explore the relationships between the Indigenous Mexican populations and the ancient samples, we estimated a

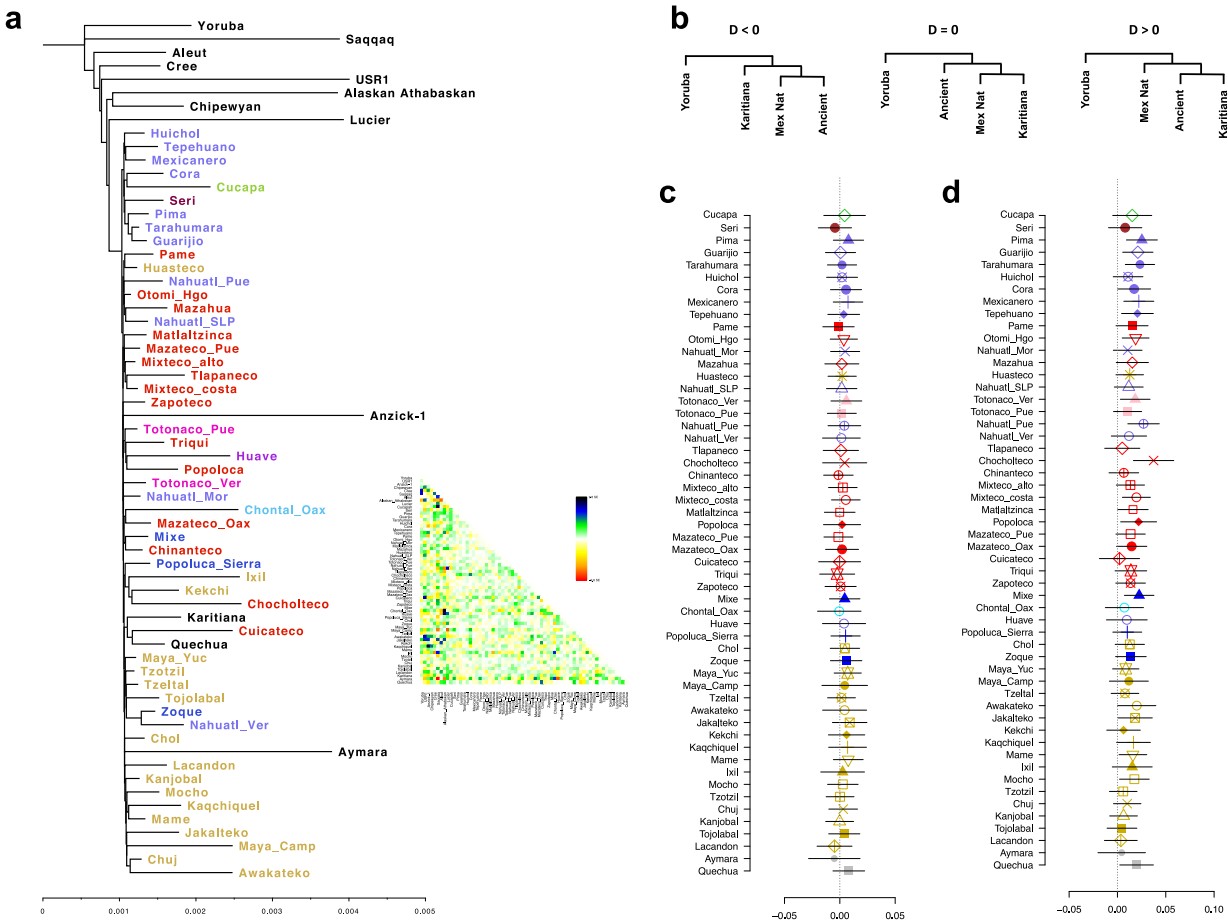

**Fig. 5 Genetic relationship of Mexican Native populations with the ancient genomes of USR1 and Anzick-1. a** Maximum-likelihood tree inferred from allele frequency, residual plot from the maximum-likelihood tree is shown. **b** Possible tree topologies resolved by *D*-statistics. **c** *D*-statistic in the form of *D*(Yoruba, USR1; Mex Nat, Karitiana) shows that all Mexican populations are related to the USR1 genome. **d** *D*-statistic in the form of *D*(Yoruba, Ancient; Mex Nat, Karitiana). Error bars in **c**, **d** represent 3 standard errors estimated by a weighted block jackknife. Mex Nat Mexican Native population.

*D*-statistic in the form of *D*(Yoruba, Ancient; Mexican Native population, South American), where the ancient samples were either USR1 or Anzick-1, and the South American samples were either Karitiana or Aymara. We also included the Tzotzil population as an internal control. When USR1 was used to represent the ancient population, we did not observe any significant deviation from zero in any of the tests (Fig. 5b and Supplementary Fig. 18a, b). However, when Anzick-1 was used in the test, we found that Anzick-1 shares more alleles with the South American population than with some of the Indigenous Mexican populations. This is particularly the case for populations from Aridoamerica, with the exception of Cucapa and Seri, as well as some Mesoamerican populations, such as Totonaco from Veracruz, Nahuatl from Puebla, Otomi from Hidalgo, Mixteco Costa, Chocholteco, Mocho, and as previously observed Mixe. Although we only found significant results ($|Z| \geq 3.2$) when Karitiana and Tzotzil were used in the comparison, we observed a similar trend when Aymara was used in the test (Fig. 5c and Supplementary Fig. 18c, d). These results suggest that some of the Indigenous populations in our data set carry ancestry from a population that split before the Anzick-1 individual. Previous studies have suggested that the Mixe carry additional ancestry from an unknown population related to the SNA/ANC-A branch that split above the Anzik-1 individual (UPopA)[33]. Our results are consistent with this observation, suggesting that other

populations from Aridoamerica and Mesoamerica may carry this ancestry (Fig. 5c and Supplementary Fig. 18c, d).

## Discussion

The findings reported here show an even more complex history of early divergences in the Mexican Indigenous populations than previously described and help to fill in the gaps regarding the human settlement of Mexico and the Americas. Taken together, these data show that the genetic structure observed in present-day Indigenous Mexican populations reflects complex demographic, cultural, and geographic events, and suggests that there may have been an overlap in the timing of these events, such as the spread of farming and cultural diversification in the differentiation of Aridoamerican/Mesoamerican populations.

The influence of the geographic location in the differentiation of Mexican Indigenous populations was observed in the PCA, which defined three clusters coincident with the North, Center/South, and Southeast regions of the Mexican territory, indicating that the geographic distribution influences the genetic structure in Indigenous populations (Fig. 1b, c). This was also supported by the correlation observed between the longitude and latitude and the PC1 (Supplementary Fig. 3a, b) and the Mantel test based on $F_{ST}$ and geographic distances, consistent with an isolation by distance model (Supplementary Fig. 3c). Moreover, $F_{ST}$-based NJ

tree identified five major regions in which Mexican Indigenous populations can be clustered (Fig. 2a).

This geographic pattern was also observed in the admixture analyses in which five of the nine identified components were observed in populations located in northern Mexican populations, corresponding to Aridoamerica, encompassing a semiarid area at the North of Mexico, whereas the sixth was present in the Lacandon ethnic group located in the Chiapas jungle in Southeastern Mexico. Both regions could act as geographic barriers, favoring the isolation of these populations and limiting the gene flow to contribute to the observed genetic structure. This was also observed in the two components detected in the Oto-mangue populations, in which the ethnic groups located around the Neovolcanic axis at the central part of Mexico, exhibited similar genetic structure (Fig. 2c, blue component), which was different from those Oto-mangues inhabiting the state of Oaxaca (Fig. 2c, red component). The ninth component was mainly observed in populations from the Mayan linguistic family. Otherwise, we observed ROH segments longer than 8 Mb in all studied populations (except for four populations), which indicates isolation episodes (Supplementary Fig. 10). Altogether, these results support previous hypotheses[2,34,35] suggesting that the geographic barriers observed in the Mexican territory have played a major role in shaping the observed patterns of genetic structure in present-day Indigenous populations. Otherwise, genetic-geographic correlations may reflect ancestral relationships and initial settlement patterns in which closely related people settled in the same geographic areas, and their descendants persisted in these regions until today. In addition, these correlations can reflect structured patterns of interaction and genetic exchange, leading to gene flow between groups in proximity with one another, possibly reflecting the influence of alliances and migrations that occurred during different periods in the history of Mesoamerica[36–38].

Cultural histories in different regions of Mexico may have also contributed to the observed genetic patterns. In this case, we observed a decline in the $N_e$ of all tested populations between 15–30 generations ago (Fig. 3a and Supplementary Fig. 8), followed by an expansion. Although our results contrast studies investigating the population history of Native Americans through admixed populations from America[21], our analyses are in agreement with recent studies modeling demographic reconstructions through time based on mitochondrial and whole exome data from ancient and modern Native American samples, which also shown a reduction in $N_e$ in recent generations[20,39]. In most cases, the timing of the observed bottlenecks corresponds with the beginning of the European colonization of the Americas (Supplementary Fig. 7) and is consistent with prior studies on the impact of settler colonialism on Indigenous communities[20,39] and historical records that provide insights into the demographic processes in specific regions[40–42]. Some authors have suggested that the total Native American population decreased by more than 90% at this time[40–42]. Overall, our findings show, for the first time, that the strength and timing of the contraction observed in Indigenous populations were not localized to a particular population but, instead, has been widespread in all tested populations, and in some cases, they took place before the European contact.

On the other hand, the estimated $T$ of 3.96–9.47 ka ago and 4.84 to 10.15 ka ago between populations from Aridoamerica and Mesoamerica (Fig. 3c, d) overlaps with the beginning and establishment of agriculture[33,36,43–46]. In this sense, though populations in Aridoamerica maintained the hunter–gatherer lifestyle over the centuries, the domestication of plants and animals in Mesoamerica as early as 7 ka ago caused a transition from Paleo-Indian hunter–gatherer tribal groups to the organization of sedentary agricultural villages[33,36,43–46]. This subsequently gave rise to the earliest complex civilizations in Mexico[36,43]. These results agree with a recent study of 76 masked exomes from Native American populations from Mexico that found a similar $T$ between populations from northern and central/southern Mexico[22]. Although these analyses cannot directly link these series of cultural developments with the genetic differentiation of the populations from the two regions, they suggest that these series of cultural events contributed to the genetic structure of the pre-Hispanic populations from Mexico. Nevertheless, the assumption of a clean split between populations from Aridoamerica and Mesoamerica could underestimate the observed $T$ due to the relatively recent gene flow between populations from both cultural areas.

IBD networks could reflect spatiotemporal dynamics between populations from different regions. Network visualization by intermediate track sizes suggests that such interregional movements occurred around 500–1500 years ago or more[27], such as revealed between Tarahumara and Guarijio corresponding to the North of Mexico and Mayan groups from the Southeast region, with Matlaltzinca in the Center or Triqui from the South region (Fig. 4 and Supplementary Fig. 11a). Historically, these findings could be supported by recent studies that have provided evidence that trade, political relationships, and local sociocultural histories have shaped the demographic histories and migration patterns, mainly in the classic and post-classic period in northern, western, and central Mexico, influencing gene flow patterns among populations and the population structures[36,47–51]. Meanwhile, recent connections around 0–500 years ago between indigenous populations from different regions were also suggested by the IBD network, through the large track size analysis (Supplementary Fig. 11b). Dynamic movements have been largely observed between indigenous populations in recent times. An example of this occurred during The Porfiriato, a dictatorial era in Mexico during 1876–1911AD, when some ethnic groups from the North were forced to work in the Southeast of the country, especially in the Mayan region[52]. Nevertheless, we should be cautious in our interpretation of historic events associated with these IBD patterns, because the time frames were inferred in European populations and could be different in other populations such as Native Americans[27]. Further studies are still needed to clarify the contribution of cultural traditions and transitions may have had on the genetic structure of present-day Indigenous peoples inhabiting the Mexican territory.

On the other hand, studying ancient genomes has been helpful to elucidate ancient population relationships and migration patterns. In the last few decades, an increasing body of evidence has suggested that the peopling of the Americas brought at least four distinct streams of migration from Asia, beginning as early as ~23 ka ago[3,8,53,54]. The main contribution has been suggested to come from an ancient population that occupied Beringia for several thousand years before moving into North and South America approximately 16 ka ago[1,4,31,32,55,56]. Recently, this ancestral population was demonstrated to be related to a lineage found in the terminal Pleistocene in Alaska (represented by the Upward Sun River burial or the USR1 individual's genome)[29], and that a split ~15 ka years ago in the far North led to the northern (NNA/ANC-B) and southern (SNA/ANC-A) population branches in the Americas[29]. The SNA/ANC-A branch includes the ancestors of the Clovis child (Anzick-1) and all Indigenous peoples from Mexico to South America[28]. Nevertheless, the complex genetic structure observed in Mexican Native populations in this study, as well as the intricate series of migrations into South America identified in two other recent studies[57,58], suggest that the history and ancestry of Indigenous populations in Mexico may be more complex than reported.

Herein, the comparison of modern Indigenous populations from Mexico with the ancient genomes from USR1, Lucier, Athabaskan, and Anzick-1 shows that Native Americans were derived from a common ancestor related to a population closer to USR1, consistent with the First American dispersal model[8,29], and that the Mexican Indigenous populations experienced one of the following scenarios: (1) a split between Aridoamerica and Mesoamerica occurred prior to the split between Mesoamerica and Anzick-1 with posterior gene flow between the Aridoamerican and Mesoamerican populations, or (2) a split between Aridoamerica and Mesoamerica occurred after they split from the Anzick-1, with Aridoamerican populations carrying gene flow from a population that split above the population represented by Anzick-1 and below the NNA/ANC-B branch (Fig. 5 and Supplementary Fig. 18). Moreover, the IBD network analysis and the TreeMix admixture graph allowing 20 migration edges inferred multiple waves of gene flow from Mesoamerican to Aridoamerican populations and vice versa, which is consistent with the first scenario (Fig. 4 and Supplementary Figs. 11, 14).

In addition, adaptation to diverse environments and subsistence lifestyles may have resulted in the selection of new alleles in present-day Native American populations[59]. These possibilities still need to be explored further, but it is likely that they, along with genetic drift and complex demographic histories, contributed to remarkable changes in the genomic patterns of Indigenous populations in Mexico.

A limitation of the present study is that genetic data were obtained using a commercial microarray that is not designed for Native American populations, which could hinder the resolution of the inferences. This also highlights the need to study these populations in greater depth via whole-genome sequencing, which will allow to identify the complete spectrum of genetic variation in Native American populations and delve into the demographic histories of these populations.

## Methods

**Samples and data handling**. The samples included in this study belong to the MAIS cohort[9–11,60] collected between 2012 and 2018. The MAIS cohort recruited genomic and clinical data from 77 indigenous communities from 60 different ethnic groups for a total of 3200 individuals[9–11,60]. All individuals were self-recognized as Indigenous members of an ethnic group and had parents and grandparents born in the same community. All participants provided informed written consent. For some of them, informed consent was translated into their native language, and some individuals signed with their fingerprints. Genomic DNA was extracted from whole blood using a commercial kit (Gentra Puregene, Qiagen Systems, Inc., Valencia, CA, USA). From this cohort, a total of 716 unrelated individuals belonging to 71 Indigenous communities representing 60 ethnic groups from 10 linguistic families were selected for genome-wide genotyping based on the availability of samples (when possible we selected at least 10 members per ethnic group, Supplementary Table 1). From this group, 644 samples were genotyped using the Affymetrix Human 6.0 array, and 72 samples were genotyped using the Illumina OMNI 2.5 array.

This study was designed in accordance with the Declaration of Helsinki and approved by the Research, Ethics, and Biosafety Human Committees of the Instituto Nacional de Medicina Genómica (INMEGEN) in Mexico City (protocol number 31/2011/I) with the support of the National Commission for the Development of Indigenous Communities (CDI, from the Spanish Comisión Nacional para el Desarrollo de Pueblos Indígenas) and with the agreement of the Indigenous leaders from each community. All participants provided written informed consent, and authorities or community leaders participated as translators when necessary.

To perform our estimations, we generated several data sets merging our genotype data with those previously published for several worldwide populations and modern and ancient Native American individuals as follows. For data generated using only an SNV array, we performed the data handling and quality control procedures in Plink v1.9[61]. Each data set was processed individually, including per marker and per sample examinations. We removed SNVs with genotyping rates <98% and those with a minor allele frequency of 1%, and then removed mitochondrial and sex chromosome SNVs. Finally, we excluded individuals with missing rates >3% and with discordant gender information.

**Mexican Natives modern populations data set**. First, we merged our 716 Mexican Indigenous samples with worldwide populations reported by Reich et al.[8], Moreno-Estrada et al.[2], and Silva-Zolezzi et al.[12]. After QC in each data set, all data were merged using the mergeit function of the Eigensoft v5.0. Software[13]. After merging, SNVs with highly discordant allele frequencies ($p < 0.001$) between the same populations in different data sets or strand bias were excluded. A total of 3490 individuals from 219 populations and 61,393 autosomal SNVs were included. We used this data set to identify the proportion of admixture with other worldwide populations in the Indigenous samples from Mexico. We ran PCA[13] analyses using default parameters and admixture analyses assuming $K = 4$ clusters, including cross-validation error estimation, a block relaxation algorithm as the optimization method, and 100 replicates. Finally, the run with the highest likelihood was selected.

**Mexican Indigenous data set**. This data set was composed of Mexican Indigenous belonging to the MAIS cohort ($n = 716$) and those previously reported by Reich et al.[8] ($n = 57$), Moreno-Estrada et al.[2] ($n = 350$), and Silva-Zolezzi et al.[12] ($n = 86$), yielded a total of 1209 samples from 60 different Mexican ethnic groups (Fig. 1a and Supplementary Table 1), representing all linguistic families except for Kickapoo. After QC in each data set and before merging, SNVs with highly discordant allele frequencies ($p < 0.001$) between the same populations in different data sets or strand bias were excluded. Finally, samples were merged and the same QC was repeated for the whole data set, including the removal of first- and second-degree relatives, yielding an intersect of 1086 individuals and 61,393 autosomal SNVs (Supplementary Table 1).

**Mexican Indigenous masked data sets**. The masked data set was generated as follows. First, we generated a reference population panel composed of 50 Native Americans from the MAIS cohort without evidence of recent admixture with continental groups (Africans and European populations) identified in the admixture $K = 4$ analyses, 50 Europeans, and 50 Africans derived from the 1000 Genomes Project Phase 3[62]. Except for the Reich data, which were previously masked[8], the reference data set was merged individually with each data set of Mexican Indigenous populations, yielding an SNV intersect of 548,310 for the MAIS cohort genotyped with Affymetrix Genome-Wide Human SNP Array 6.0; 214,968 for the MAIS cohort genotyped with the Illumina HumanOmni 2.5-4v1_B SNP array; 505,024 for the data set reported by Moreno-Estrada et al.[2], and 303,609 for the data set reported by Silva-Zolezzi et al.[12]. Prior to performing local ancestry, each set was phased using SHAPEIT v2.17[63] with default parameters. The local ancestry estimation was performed using RFMix v2[15] with two EM iterations and a forward-backward threshold of 0.9. Non-homozygous native tracts identified by local ancestry estimation were then masked in each individual by setting the genotypes to missing, admitting a maximum threshold of 40% of the genome masked for each sample. After masking, we merged all data sets, including the masked set from Reich et al.[8], yielding a total of 3490 individuals from 218 populations and 61,393 autosomal SNVs. To test the accuracy of masking, we ran admixture analysis in this data set with $K = 4$ with the previously specified parameters (Supplementary Fig. 2, lower panel).

Finally, we generated two masked data sets of indigenous populations. The first one was generated by extracting the Native American individuals from the masked data set used to verify the masking accuracy. This data set includes 1086 individuals from 60 Native American populations and 61,393 autosomal SNVs. The second masked data set included the masked data sets from the MAIS cohort and Moreno-Estrada et al.[2], both genotyped using the Affymetrix Genome-Wide Human SNP Array 6.0, yielding a total of 996 individuals from 60 populations and 504,581 autosomal SNVs (Supplementary Table 1).

**Ancient and modern Native American data sets**. Fastq files from publicly available ancient genomes from the Americas and present-day South Americans were downloaded from their respective repositories[3,28–31]. Adapter sequences, were removed with AdapterRemoval v2.1[64]. The reads were mapped to the human reference genome build 37 using bwa aln v0.6.2-r126[65] with disabled seeding (-l parameter)[66]. PCR duplicates were removed using picard-tools MarkDuplicates[67], and local realignment was carried out using the GATK best practices guide[68,69]. Diploid genotypes were called using HaplotypeCaller from GATK v4.0[68,69].

The gvcf files generated for each ancient genome were called together using the CombineGVCFs command of GATK, converted to plink format using VCFtools[68,69], and then converted to eigenstrat format using the convertf program from Eigensoft v5.0[13]. Finally, these data were merged with the array data from the 325 individuals from the MAIS cohort with a Native American ancestry >99 (Supplementary Table 5) using the mergeit function from Eigensoft v5.0. Software[13], yielding a total of 111,586 autosomal SNVs.

**Population structure analysis**. The genetic structure of the Mexican Indigenous population was determined in the masked data set including only Indigenous populations from Mexico using both PCA[13] and admixture[14] analyses. For admixture analysis, we simulated $K = 2$–16 clusters including cross-validation error estimations and the block relaxation algorithm as the optimization method. For

each $K$, we ran 100 replicates and selected the run with the highest likelihood. Finally, we compared the cross-validation value from each estimation to determine the $K$ with the lowest cross-validation error value.

**$F_{ST}$ calculation and the neighbor-joining tree**. The $F_{ST}$ matrix between population pairs was assessed in the masked data set by the smartpca function included in Eigensoft v5.0. Software[13]. Based on this matrix, a NJ tree was constructed using R package APE v5.1[70]. To evaluate the tree topology, a set of 100 bootstrap simulations was performed. To generate bootstrap replicates of the NJ tree, we randomly removed 100 times 1% of the total SNVs of the masked data set. With each set of remaining variants, we generated 100 independent pairwise-$F_{ST}$ matrices to generate a new NJ tree from each one (replicates) in R package APE v5.1. The consensus tree was generated under DendroPy v4.0.0[70] using the whole data NJ tree to maintain the topology, and each replicate was used to obtain the bootstrap values.

**Inference of isolation by distance**. To test whether isolation by distance explained part of the genetic diversity observed in Native populations from Mexico, we performed a Mantel test for a matrix of pairwise-$F_{ST}$ statistics and geographic distances between populations in R package adegenet v2.1.1[71].

**Effective population size calculations**. The effective population sizes of Indigenous Mexican populations were estimated in a set of 48 selected populations from the second masked data set (Supplementary Table 2). The populations were chosen if the genotyping platform was Human Affymetrix array 6.0 and if there was a minimum sample size of 10. This ensured enough SNV density and the minimum sample size. We used two different software for our estimations. First, we determined the $N_e$ across time from the LD analysis based on the method reported by McEvoy et al.[16] in R package NeON[17] with default parameters. To estimate more recent demographic history, we estimated $N_e$ by identifying IBD tracks based on the pipeline reported by Browning et al.[21]. Briefly, the data sets were phased together using Beagle 5.1[72]. IBD segments in the data from each selected population were detected by IBDseq[19]. We then used IBDNe[18] for nonparametric estimation of the recent demographic history from IBD segments with the default parameters and a minimum IBD segment length of 2 centiMorgan (cM) to estimate the $N_e$ across generations.

**Divergence time estimations**. We estimated the time of divergence between pairs of populations ($T$) in generations in the 48 selected populations mentioned in the previous section (Supplementary Table 2) using the following formula as reported by McEvoy[16]: $2N_e F_{ST}$, where $N_e$ is the long-term effective population size estimated in each population and $F_{ST}$ is the genetic distance between pairs of populations. The long-term $N_e$ was estimated based on LD patterns using the method reported by McEvoy et al.[16] in R package NeON[17]. Divergence times between pairs of populations were determined using the *Tdverg* function from NeON package[17].

**Runs of homozygosity**. ROH were calculated by Plink v1.9 software[61] in the set of 48 selected populations using the following parameters: SNV density 50 SNPs per window, minimum ROH length 1 Mb, two missing genotypes per window allowed, one heterozygote per window allowed, sliding window of 1 Mb, and minor allele frequency >5%. To obtain the proportion of the genome of an individual in ROH, we divided the total base pairs identified in ROH by the total approximate length of the autosomes (2.8 Gb).

**IBD analysis**. Hap-IBD v1.0[26] was used to detect IBD segments between individuals with >99% Native American ancestry (Supplementary Table 4) in order to avoid spurious results due to missing data. Phasing was performed using SHAPEIT v2.r837[73] with default parameter settings. Hap-IBD reports all IBD segments identified by each pair of individuals. In order to analyze results, data were divided by grouping the individuals according to their community or region of origin. As described in Ioannidis et al.[74], we assessed the probability that an individual selected at random from population A shares an IBD track greater than 7 cM (in order to avoid spurious matches) with an individual selected at random from population B. Therefore, we divided the total number of individual pairs connected by more than 7 cM of IBD by the total number of possible individual pairs, for each pair of indigenous groups analyzed. Following the definition of the five geographical regions identified in the NJ tree and previously described in Contreras-Cubas et al[9]. IBD networks were generated in R with function ggnetworkmap from the GGally v2.1.2 library. Vertices are located on geographical sampling coordinates from each population. Connections between groups have a width proportional to the probability of any individual from population A sharing at least one IBD segment with any individual from population B. Five IBD networks were generated, where each figure focuses on the connections from populations within a given geographical region: North, Northwest, Central-east, South, and Southeast (Fig. 4). Edges between populations are color-coded, where blue, violet, orange, green and yellow represent connections within or between northern, northwestern, central-eastern, southern or southeastern populations, respectively. Supplementary

Data 3–7, presents the numerical value for the probabilities of detecting an IBD track between pairs of populations for the five networks.

In order to gain a spatiotemporal resolution of the connectivity between indigenous populations of different regions, and IBD analysis was made using tracks of intermediate (5–10 cM) or large (>10 cM) size, which have been reported to originate at different time points in the past. This analysis was made incorporating the populations at the regional level to display with a more detailed resolution, the difference in connectivity between different regions (Supplementary Fig. 11).

**F-statistics**. We used the f-statistic framework to explore the relationship between USR1 and Anzick-1 ancient genomes and modern Mexican Native populations. The outgroup f3-statistic and $D$-statistic tests were computed in Admixtools v4.1 using the estimators described in Patterson et al.[75], and standard errors were obtained using a block jackknife procedure over 5 Mb blocks in the genome.

**Maximum-likelihood tree**. A maximum-likelihood tree was constructed using TreeMix v1.13[76] based on the Ancient and Modern Native American data sets. The tree was rooted using the Yoruba population as an outgroup, and standard errors were estimated using blocks of 50 SNVs.

Alternatively, we tested the presence of gene flow between Mexican Indigenous populations assuming 1–20 migration edges with TreeMix using the previously generated maximum-likelihood tree to maintain the topology of each tree and USR1 genome as an outgroup.

**Reporting summary**. Further information on research design is available in the Nature Research Reporting Summary linked to this article.

## Data availability

The whole-genotype data from the 716 Indigenous individuals from the MAIS cohort is available under restricted access to protect the privacy of the participants and in alignment with the Institutional Review Board approval and the individual informed consents forms. Access can be obtained by researchers at research institutions through a data-access agreement. Please contact L.O. (lorozco@inmegen.gob.mx) or H.G.-O. (hgarcia@inmegen.gob.mx).

## Code availability

Details about methods, software, and pipelines used are included in the "Methods" section.

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

## Acknowledgements

We thank the volunteers who donated their DNA samples for the realization of this project. This study was supported by the Consejo Nacional de Ciencia y Tecnología (http://www.conacyt.mx/) grant S008-2014-1 No. 233970 (L.O.). The funder had no role in the study design, data collection and analysis, decision to publish, or preparation of the manuscript. P.B. and M.F.-H. were supported by Consejo Nacional de Ciencia y Tecnología (http://www.conacyt.mx/) fellowships (nos. 607882 and 596612, respectively). This article is dedicated to the memory of our colleagues José Concepción Jiménez-López and José Sánchez-Corona.

## Author contributions

Steering Committee: L.O., H.G.-O. and A.M.-H. Analysis of genetic data: H.G.-O., F.B.-O., C.C.-C., M.A.C.-S., P.B., M.F.-H., F.S.-Q., A.B. and A.W.R. Sample collection: L.O., A.M.-H., E.M.-C., H.G.-O., F.B.-O., C.C.-C., M.A.C.-S., M.F.-H., P.B., M.S.-M., J.S.-C., A.R.-M., S.E.F.-M., R.O.-L., C.G.-H., A.G.G.-Z., E.J.C., F.C.-C., B.P.L.-R., G.S.-M., F.L.-B., I.C.-A., J.L.J.-R., H.R.-V., R.V.-C. and M.M.-M. Data generation: L.O., H.G.-O., A.M.-H., F.B.-O., C.C.-C., M.A.C.-S. and R.M.-E. Manuscript writing: L.O., H.G.-O., F.B.-O., C.C.-C., A.M.-H., M.A.C.-S. and M.F.-H. Critical review of the manuscript: D.A.B., M.S., M.S.-M., J.S.-C., A.R.-M., S.E.F.-M., R.O.-L., A.G.G.-Z., E.J.C., F.C.-C., J.C.J.-L., E.G.S.-B., Y.S.-A., G.L.-R., E.M, A.W.R., H.R.-V., A.D.K., M.C.A.-P., J.R.-M. and F.S.-Q.

## Competing interests

The authors declare no competing interests.

## Additional information

¹Instituto Nacional de Medicina Genómica, Tlalpan, Mexico City, Mexico. ²Department of Anthropology and Institute for Systems Genomics, University of Connecticut, Storrs, CT, USA. ³Department of Anthropology, University of Montana, Missoula, MT, USA. ⁴División de Medicina Molecular, Centro de Investigación Biomédica de Occidente, Instituto Mexicano del Seguro Social (IMSS), Guadalajara, Mexico. ⁵Tecnologico de Monterrey, Escuela de Medicina y Ciencias de la Salud. Universidad Autonoma de Nuevo Leon, Centro de Investigacion y Desarrollo en Ciencias de la Salud, Monterrey, Mexico. ⁶Department of Anthropology, Baylor University, Waco, TX, USA. ⁷Harvard Medical School, Joslin Diabetes Center, Boston, MA, USA. ⁸Instituto Politécnico Nacional, CIIDIR-Durango, Durango, Mexico. ⁹Departamento de Farmacobiología, Centro Universitario de Ciencias Exactas e Ingenierías, Universidad de Guadalajara, Guadalajara, Mexico. ¹⁰Dirección de Antropología Física, Instituto Nacional de Antropología e Historia, Museo Nacional de Antropología, Mexico City, Mexico. ¹¹Unidad Académica de Ciencias Químicas, Universidad Autónoma de Zacatecas, Zacatecas, Mexico. ¹²Section for Evolutionary Genomics, The GLOBE Institute, The University of Copenhagen, Øster Farimagsgade 5A, 1352 Copenhagen, Denmark. ¹³Universidad Michoacana de San Nicolás de Hidalgo, Michoacán, Mexico. ¹⁴Instituto de Investigación en Genética Molecular, Universidad de Guadalajara Ocotlán, Jalisco, Mexico. ¹⁵These authors contributed equally: Humberto García-Ortiz, Francisco Barajas-Olmos, Cecilia Contreras-Cubas. ¹⁶Deceased: José Concepción Jiménez-López, José Sánchez-Corona. ✉email: hgarcia@inmegen.gob.mx; lorozco@inmegen.gob.mx

