## [Peer Review File · Nature Communications]

The genomic landscape of Mexican Indigenous populations brings insights into the peopling of the AmericasReviewers' Comments:

Reviewer #1:

Remarks to the Author:

García-Ortiz et al. present SNP array data from 706 Indigenous Mexican individuals from 60 out of the 68 recognised ethnic groups in Mexico. By analysing these data jointly with publicly available data from other Indigenous groups in Mexico and the Americas, they explore the genetic structure of Indigenous Mexicans in the context of linguistics and, in particular, geography. The authors find that groups that are closely related in terms of culture and language often form genetic clusters. However, geography appears to be the major driver of genetic structure in Mexico. Based on estimated demographic parameters such as N_e and pairwise population split times, García-Ortiz et al. explore the demographic history of Indigenous Mexicans and conclude that Aridoamerican (AA) and Mesoamerican (MA) populations diverged from each other early on, presumably as a consequence of the adoption of agriculture in MA.

Given the claims presented in this article and especially, the extent of the sample, I consider that this is a valuable work that will be appealing to the broad audience of Nature Communications. However, I would like the authors to include the following suggestions in a revised version.

1. I would like the authors to include a more detailed description of the datasets that were used for each analysis. A paragraph or a table summarising the populations (whether they were masked or not) and number of SNPs per analysis should make it easier for the reader to follow the text. In this regard, please be more specific about what the values in Fig3a and Fig4d represent.
2. Since the manuscript stresses the importance of geography in the determination of the genetic structure of Indigenous Mexicans, Fig1 could be more informative. Please include the the major/main geographic features in Mexico, e.g., mountain ranges. Adding these should make the manuscript more accessible to readers that are not familiar with Mexican geography. Most importantly, highlight AA and MA.
3. There are two places in the manuscript where the authors could better discuss the potential effect of gene flow on their inferences.
 - A clean split was assumed in order to estimate pairwise population divergence times. Given that estimating the demographic parameters for an isolation with migration model using SNP array data of this kind would be a really demanding task, I would encourage the authors to openly discuss (additional to the statement in l274) this major limitation.
 - D-statistics of the form $D(\text{Han}, \text{Anzick1}; \text{Karitiana}, \text{Indigenous Mexican})$ are used to claim that AA diverged prior to the Anzick1-MA split. However, this conclusion is not be consistent with the <10ka AA-MA divergence times reported in the paper, in light of the age of Anzick1 (~12.8ka). If Anzick1 forms a clade with MA to the exclusion of AA, then the MA-AA split should predate 12.8ka. Gene flow between AA and a Native American outgroup or between AA and MA excluding the population represented by Anzick1 could be discussed. These two possibilities could be explored through D-statistics of the form $D(\text{Outgroup}, \text{NNA/ANC-B}; \text{AA}, \text{MA})$, $D(\text{Outgroup}, \text{AA}; \text{Anzick1}, \text{MA})$, and $D(\text{Outgroup}, \text{MA}; \text{NNA/ANC-B}, \text{AA})$. If the latter yields non-0 results, it would be interesting to show a plot of $T(\text{AA}, \text{MA})$ as a function of $D(\text{Outgroup}, \text{MA}; \text{NNA/ANC-B}, \text{AA})$ for different (AA,MA) pairs.
4. Following the previous comment, it has been shown that some NNA/ANC-B populations have East Asian/Siberian-related ancestry. Therefore, I suggest the usage of an African population in their treemix and D-statistics analyses.
5. For the paragraph starting in l222. on N_e , I suggest the authors include a couple of sentences with a disclaimer on how N_e differs from actual census sizes. This could be helpful for the broad readership.
6. While it is implicit, it is not clear what the authors mean by RoH coefficients. It would be more

informative if the authors report more 'standard' RoH metrics such as the proportion of the genome found in RoHs (shown as a boxplot per population).

In addition, a size analysis of the RoHs per population following Pemberton et al. 2012 (10.1016/j.ajhg.2012.06.014) could be helpful in hypothesising when the isolation took place.

I would have expected a population as isolated as the Tarahumara to carry many long RoHs. Could the authors speculate why this was not the case for the Tarahumara and the other 'divergent populations'?

7. In l248, the authors talk about the role of geography in the observed genetic patterns. An example would be helpful here, together with the inclusion of geographic features in Fig1.

8. For the USR1 and Anzick1 genomes, it is not clear how the merging was done. Please include a Methods section with information on whether a genotyping or a random allele were sampling strategy was used. Were any bases trimmed from the ends of the reads?

9. The f3-statistics analysis could be presented differently. Given that the intended message is that Indigenous Mexicans are similar to other Native American populations, it would be more informative to compute f3 based on the full worldwide dataset shown in EDFig1. Include figure/table showing that Indigenous Mexicans and other Native Americans have similar values.

10. For the treemix graphs, the placement of Cree and Ojibwa with respect to USR1 and Arctic populations is a bit unexpected. Using CHB as an outgroup could be the cause due to the East Asian/Siberian-related ancestry in some northern populations (see 4.).

The method description for this analysis states that only samples with 99.99% Native American genetic ancestry were used. However, the Cree, Ojibwa and Arctic populations from Reich et al. 2012 have been shown to be heavily admixed. Were these samples treated in a particular way? If not, could Eurasian admixture explain the graphs in EDFig8?

If the 60k site set was used for this analysis, 500 SNP-blocks would result in two few blocks. For f-statistics, ~5Mb-blocks are often used.

Finally, did the authors perform multiple optimisations with different starting points? If so, please specify. The same comment goes for the ADMIXTURE analysis.

11. In the local ancestry deconvolution methods section, please specify the masking scheme. Did the authors mask all sites that were not 'homozygous' Native American?

Fig3.

- In D, please use the color scheme in Fig2

Fig4.

- Please define what the error bars in b,c represent
- Is there a particular reason for using the term 'Outgroup D'
- Please use color scheme in Fig2

l74-75. 'ancient and more recent events' is too vague

l76. Please describe the number of new genotyped individuals, not the merged dataset.

l84. 'Mexican/South American indigenous peoples are derived from USR1' is a bit misleading. 'USR1 is an outgroup to' could be easier to follow.

l134. Genetic structure instead of diversity

l136. Mexican Indigenous populations

l142. It is not clear that diversity is being 'measured' here. Thus I suggest to lower the tone of this statement.

l146. Correspondence instead of correlation (unless there is a correlation score)

l165. For clarity, I suggest that the authors highlight the cited examples in Fig2a.

l174. Is there a statistical test for significance here?

l216. have shown

l518. Genetic structure instead of variation

EDFig9. Northern branch

EDFig10. North America

I hope these comments will help the authors improve the manuscript, which I expect to see published in the near future.

Reviewer #2:

Remarks to the Author:

In the paper "Landscape of genomic diversity in Mexican Indigenous populations: insights into the peopling of the Americas" García-Ortiz and colleagues examine the genomic diversity of a large cohort of indigenous Mexican populations (68 groups). Their main finding is a structure between populations from Mesoamerica and Aridoamerica which corresponds to different ecogeographic domains. The populations from Mesoamerica belong to the previously defined ancestry of Amerindian that comprises the rest of South American Native populations, and is close to the ancient remains of Anzick.

The paper is well written and the data released represents possibly the richest survey of Native American diversity for a Latin American country. The north-south structure in Mexico is relevant to understand early migration movements in the continent. Nevertheless, the analysis included in the manuscript are not very conclusive and some of the claims are not sufficiently justified. Most of the results are in fact based on descriptive statistics such as F_{ST} distances and simple calculations of divergence time from N_e and F_{ST} which do not contemplate confidence interval.

The authors should give more insights to read the genetic diversity of this impressive dataset. At the moment it is not clear what element of novelty emerges from the analysis, and the level of descriptiveness and speculation is predominant.

To understand the ancestry of the Aridoamerican populations in particular, it would be good to directly compare the data to other ancient data recently published, with a focus on North America, such as Posth et al. 2018 and Scheib et al. 2018.

To understand the timing of the north south split with better resolution I suggest to perform demographic simulations with a Bayesian framework such as in Harris et al. 2018.

The non-Native American ancestry through the samples is not sufficiently described. This would be important to understand how strong are the claims based on the Native American ancestry alone and then follow which analysis are based on the samples with the high Native American ancestry only.

Finally, the genetic structure of the country should be better contextualized against what we already know for the continent from previous genetic studies: genomic studies, but also studies based on uniparental markers or HLA types.

In particular, the evidence for a bottleneck around 20 generations ago is not conclusive. From the Supplementary Figure 6, there is evidence of a steady smooth decline for most population, and an increase in size in the last 20 generations. Only populations like LCDN, MXCN, HUI, TNEK, THRA, NAH_MOR show some kind of decline (not that drastic). Many subtle signals of bottleneck point at 25

or 30 generations ago. From this figure alone, it is not possible to point at a strong bottleneck in the historical time frame of the post-Columbian contact. Demographic testing with simulations could be an option to confirm the validity of a bottleneck in a compatible time frame.

More details comments on the manuscript are following.

Abstract: spell out USR1 for non-specialists.

Introduction:

Expand the introduction by talking about previous knowledge on the genetic structure of Mexico.
Line 115: why the most diverse population in America is in Mexico?

Results:

Mention how many SNPs are included in the analysis after merging the data. In general, mention if the whole set of populations is included in each analysis, or which part of it.

Line 141: which hypothesis? From which evidence?

142: why are we talking about geography in shaping diversity? We only see three blocks corresponding to three major regions, but the geographic differences of this structure are not explained.

The F_{ST} based NJ tree is too simplistic to draw major conclusions based on the linguistic affiliation of the samples. The methods say that bootstrap was performed: where can we see this? How much is this "high clustering" of linguistic families? From the tree in particular it is impossible to see if there was any linguistic radiation, and reflect on the demographic spread hypothesis of the Uto-Aztecan suggested by Diamond and Bellwood. More linguistic reference and hypothesis should be introduced to make proper claims on linguistic-genetic relationships. It should be checked if in some cases populations who speak a language of the same family cluster together despite being geographically distant, or if a population is closer to a distant one (but linguistically related) than to a non-linguistically related neighbor.

It would be good to discuss a bit more the groups coming from ADMIXTURE analysis. is there any cultural, regional or historical region for these clusters?

Line 196-199: the difference between the second and third hypothesis is not clear. They both concern relatively recent gene-flow.

The difference between the two methods to detect N_e should be better explained. Which method was used to calculate long-term N_e ? As said before, the descriptive interpretation of these plot is not corresponding to the apparently clear pattenr described. Definitely not with a reduction of 90% of the population as referred to historical records.

Line 218-220: not clear which changes in allele frequencies and how could be linked to current genetic diversity.

Line 222: how was this result obtained?

About the ROH analysis: what are the numbers in Supplementary Table 4? Is it the length? Is it proportion? How long are these long ROHs? This result should be displayed with a clearer output. See for example Schroeder et al. 2018 (PNAS). At the moment it is not clear what it adds to the results.

Divergence time:

Mention how high are the highest T. is it compatible with the history of the region?

Line 248: which mountains? Where are those? Better explain about potential geographic barriers. Would it be relevant to show them on a map (like fig.1)? is the distribution of pairwise divergent time coherent between the grouping proposed in Fig 3d? it is difficult to see patterns from Fig 3c. Because the method relies solely on F_{ST} distances and N_e and it is not adjusted for uncertainty and pairwise migration, and no confidence interval is reported, it would be important to see how robust is the

pattern overall and how it could be driven by drift and contact. For example, it is not clear how the values in line 264 are calculated.

The divergence dates should be discussed against other results for the Americas, such as Harris et al. 2018, for example.

Line 282: be specific about these streams and the previous literature that proposed them.

Line 294-295: this statement is particularly vague.

Line 297: mention how many SNPs are left after merging with ancient DNA data.

Line 319-324: this statement is particularly vague.

Discussion:

In general, the discussion is not adding much perspective to the previously discussed results.

Line 340: how is the complex geography contributing to population diversity? In which sense is the geography of Mexico complex?

Line 350: the link is not clear, it seems very speculative.

Line 360-363: provide references and explain this statement.

The last part of the discussion is a repetition of concepts already expressed elsewhere in the manuscript and broad speculations.

Methods:

The use of linguistic labels should be better justified. For example, the Yuto-Nahua is commonly referred as Uto-Atzecan, while the Oto-Mange is referred as Oto-Manguéan.

Explain which samples are newly published and separate them from published data. Mention which Native American published data was involved, and which data from other continents and why. For example, it is not clear which reference was used for non-Native American admixture.

The order of the methods does not follow the order of the results. The global ancestry estimation should be discussed first in the results, as well.

What is the difference between the analysis mentioned in line 502 and the one in line 514?

Which N_e was used to calculate divergence time?

Is the Treemix analysis useful at all? It is mentioned solely to confirm a basal position for USR1.

Supplementary figure 9 with migrations is not really discussed in the text.

Main figures:

Fig. 2: which order are the populations in the heatmap? Can the order be the same of the populations in Fig1c Legend? So it is easier to track down similarities.

What is the color coding corresponding to in Fig. 3b, 4b and 4c? can the order of the populations be consistent?

Reviewer #3:

Remarks to the Author:

The manuscript "Landscape of genomic diversity in Mexican Indigenous populations: insights into the peopling of the Americas" presents population genetics analyses based on genotyping array data from newly-generated as well as previously published data belonging to Indigenous populations from Mexico. Of value, the authors have made an important contribution by expanding the catalogue of Mexican genetic variation to more linguistic groups. The results from the analyses carried out in this study, have few new implications regarding the peopling of the Americas. Although the work has some limitations, the study has potential as it provides newly testable hypotheses that can be addressed with less ascertained (i.e. non array) data is available. As it stands, I feel the work lacks enough

information to make it reproducible, therefore I have several minor and few major comments about the work that, if addressed I believe could improve the manuscript and be useful to the scientific community interested in the field.

General comments:

It would be more informative to have a better definition of what is Meso and What is Aridoamerica, the current description of the limits by saying "Center and South, and North" is very ambiguous, perhaps highlighting the regions in the map from Figure 1 would help. Also the authors want to mention that these regions have shifted as a result of climatic fluctuations in the last 1000 years.

I believe the authors should have a statement of the limitations of using genotype and the risk of ascertainment bias, given the arrays used were not designed with Latin American populations (let alone Native Indigenous populations) in mind. They should mention what they did to control for this in each analysis or justify why this should not be concern. Along these lines, the authors mention in several occasions the implications of their results (and previous) in light of accumulation of population-specific private alleles. However they do not acknowledge that their approach is unable to capture this.

Sometimes arguments that are more suitable for the discussion are introduced in the results, making it confusing to distinguish what is an actual results from the study and what is information just needed for discussion of results (e.g. discussion in lines 255-260 that are found in results). This is common throughout the text so I encourage the authors to revisit the manuscript and delimit better the sections.

Abstract:

"than has been recognized so far" -> "than what has been recognized so far" (though I recommend a better phrasing.

Introduction:

-The authors make the claim throughout the manuscript that this is the largest cohort of Indigenous individuals sampled to date. I believe it is important from the introduction to clarify unambiguously what is the actual contribution of new data, as opposed to previously published datasets. In line 124 they state:

"Here we report the results of a population genetic study on the largest cohort of Indigenous individuals sampled in Mexico to date. We included a total of 1,086 individuals from 60 different Mexican ethnic groups, representing all linguistic families [...] Genotypes from Mexican Indigenous populations were compared with those from 146 previously published worldwide populations^{2,8,9}."

The sentence can be misleading as it could imply that the authors have generated data from 1,086 Indigenous individuals themselves. I believe that for sake of recognizing previous work it should be stated unambiguously and from the very beginning, (i.e. here) that the data that was generated for this study comprises 633 individuals, while the remaining is from previous studies. This information is only described in the Methods section, but it should be clear from the introduction.

Results

-Line 153: don't -> do not

-It is not clear what is meant from the statement:

"While Bellwood proposed that speakers of the initial Yuto-nahua and Mayan languages spread far outward from central Mexico once they had become farmers, it is not clear that there is strong evidence for the linguistic radiations he proposed"

Specifically "it is not clear that there is strong evidence", does the authors mean it is not clear from their data? If so, is it correct to say that because a genetic signal is not observed, then a linguistic radiation likely did not take place? I would argue that language can be diffused culturally, without a strict need of gene flow, therefore I would suggest the authors to review this sentence.

-Lines 160 – 161: "They do not do so perfectly" -> It is unclear what is meant by this. What would a "perfect" clustering be?

-Line 172: "Pairwise FST comparisons identified the Tarahumara, Guarijio, and Cucapah in the North of Mexico, and those previously published like the Seri (North) and Lacandon (Southeast)² as significantly different genetically (Fig. 2b)."

It is unclear from the phrasing what are the "pairs" are in the "Pairwise FST comparisons".

-Lines 183 – 186: "Based on this K, six of these clusters were mainly observed in a single population (Seri, Tarahumara, Pima, Tepehuano, Huichol and Lacandon), while two were mainly observed in several ethnic groups inhabiting the Centre/South (here referred as multi-ethnics), and one in populations in the Southeast that are part of the Mayan linguistic family (Fig. 2c)".

As is, the sentence implies that six clusters were observed in a single population. i.e. that single population has six clusters. I only managed to understand the sentence when looking at the Figure. I believe the sentence can be rewritten to better convey the message of Figure 2c.

-Line 191: "These data, along with the Mantel test, strongly support the notion that geography has played a crucial role in shaping the genetic diversity across these populations"
Could authors add that this has already been suggested by previous genetic studies?

-In the paragraph comprising lines 191-199 the authors propose three alternatives to explain the genetic-geographic correlations. They say these:

- 1) may reflect ancestral relationships and initial settlement patterns, where closely related people settled in the same geographic areas, and their descendants persist in those regions until today.
- 2) can reflect structured patterns of interaction and genetic exchange, leading to gene flow between groups in proximity with one another.
- 3) could also reflect the influence of alliances and migrations that took place during different periods in the history of Mesoamerica.

I actually think all three are related, population continuity of people settled in the same geographic area implies gene flows between groups in proximity with one another. As to the third alternative, they mention migrations, which if long distance would then not reflect the genetic-geographic correlations, unless they mean migrations between groups in proximity, which falls back to points 1 and 2. I think authors should be clearer in discussing this point.

-"Based on historical records that provide insights into the demographic processes in specific regions, some authors have suggested that the total Native American population size in that time decreased by more than 90% at this time. In addition, recent studies based on ancient and modern Native American genomes have showed a significant reduction of ~50% in the Ne. Our results are consistent with these previous observations."

In light of previous reports, is it possible to add a % estimate of Ne reduction in the data here

analyzed?

-Can the authors lay succinctly the basic underlying principles of the IBDNe approach?

-Line 222: What is meant by “mean long-term N_e ”?

-When discussing population-specific N_e , the authors might want to consider a recent previous report (<https://www.biorxiv.org/content/biorxiv/early/2019/01/30/534818.full.pdf>) that calculated N_e in Huichol, Tarahumara, Triqui and Rarámuri (Tarahumara) and contrast to this.

-Line 239: “The most divergent populations have the highest value for T (Seri, Guarijio, Tarahumara, and Lacandon; Fig. 3d, supplementary Table 5)”

-Line 264: The divergence times are interesting. Since the authors have already calculated it, could they state estimated divergence estimates between Central/South and Southeast. Also, they might also want to consider and discuss in the context of previous estimations done in <https://www.biorxiv.org/content/biorxiv/early/2019/01/30/534818.full.pdf>, which are concordant and further support, their results.

It is not clear in respect to which other populations have the highest values.

-“Furthermore, recent evidence based on ancient DNA studies also reveals a much more complex history of migrations, divergence, and large-scale movements between North and South America, supporting our findings from the study of present-day Native population in Mexico.”

Can authors be more specific about the “complex events” that support their findings?

Discussion

-“Our results show that the genomic variation observed in Mexican Indigenous populations is the result of complex interactions between geographic, cultural, and demographic events that have shaped the genetic architecture of these populations.”

In fairness, genomic variation of all human populations is the “result of complex interactions between geographic, cultural, and demographic events”. Their results show interesting patterns regarding split times and N_e fluctuations, but really as they don’t directly assess (but rather speculate—sometimes righteously) about the role of past events, I believe the sentence above is fat stretched, even more considering that previous reports on genomic data from Indigenous individuals have already shown many of the patterns here revealed.

-“The complex geography of the Mexican territory seems to have contributed to the isolation of some groups, such as Seris and Lacandons, by limiting gene flow and favouring the enrichment of local private alleles that are rare or absent elsewhere.”

This is consistent with theory, and has been proposed already elsewhere (Moreno-Estrada 2014), but has not been really shown as part of this study as it is not possible to address with genotype data.

- “we observed that all tested populations experienced a strong bottleneck around 20 generations ago, consistent with the beginning of the European colonization and conquests that disrupted native societies and led to the demographic decline of these populations”

I think this is a very neat, although anticipated, observation. I believe it merits some discussion in the context of the paper by Browning et al.

<https://journals.plos.org/plosgenetics/article?id=10.1371/journal.pgen.1007385>

In which they do not find such a bottleneck when performing an ancestry-specific N_e estimation through time in admixed Mexican population, but they do in other American populations. Along these lines I think it is also necessary to discuss these observations and the argument made several times about the accumulation of private (potentially deleterious) alleles in the context of present-day admixed Mexican populations. In light of this, it would be very interesting also to observe the difference between N_e calculation through time in individuals that are >99.9% indigenous, and those with more admixture with and without a masking step.

-Lines 361-363: In addition, previous studies have pointed to the complex relationships ..."
Can they cite those previous studies?

-Line 365: "the comparison of modern Indigenous populations from Mexico with the ancient genomes from USR1 and Anzick-1 suggests that populations from Mexico came from the same genetic stream as the First Peoples to inhabit the Americas 8,45, but that these populations experienced an early split between the Aridoamerica and Mesoamerica/South America branches in the settlement of North America (Fig. 4)."

I think this is perhaps the most important and novel contribution of the paper, therefore I think it needs further testing and better quality discussion.

Because the data is derived from different platforms (sequencing in the ancient, versus genotyping in modern), and since it has been shown that comparing platforms/methods could introduce some bias (as more similarity between two samples/populations could be product of these being processed on the same platform). Is Karitiana sequenced data or genotyping? Authors should describe at length why they do not think this is affecting their analyses and prove it. They should also discuss how ascertainment in the genotyping arrays could be affecting their results. Furthermore, the test is made with a Han sample, I believe that a better outgroup should be a Sub-saharan African. Actually from figure 4 it is not crystal clear that all AA populations are closer to USR1 than to Anzick1. Maybe is only because of a problem representing the results in the figure (see comments specific to this figure below).

##Methods

-For each analysis please be explicit about the number of samples used (ideally referencing a table with the individuals per analysis), the exact filters, and the number of sites passing.

-The methods section needs major improvements, many details needed for reproducibility are lacking:

-Mention which local and research committees (IRBs?) approved the protocol and report the number of the project approved. This is particularly important given the work deals with very vulnerable and marginalized populations, so many ethical considerations are in place and need to have been approved by IRBs and have been consulted with local Indigenous groups.

-Line 478: "A subsample of MAIS" -> subsample of what size

-Line 492: " $p < 0.001$ " -> what test?

-More details are needed for ADMIXTURE, where there any "pruning" steps? Did they run it only once? Admixture needs to be run several times (recommended between 10 and 100) and then runs need to be condensed or the one with the best Log Likelihood be selected. There is no mention whatsoever of this.

-More details are needed for Local ancestry estimation. The authors need to explicitly say which samples were used in each of the parental populations. Was SHAPEIT2 run in the Population phasing or in the haplotype reference version? Was RFMIX run with one or several EM iterations? What

threshold was used in the forward backward values? Because masking is done by making sites missing, is there a missing threshold applied? Again, it needs to be specified how the ADMIXTURE to corroborate that the masking scheme worked was run.

-Effective size calculation: What 42 populations? Selected on the basis of what criteria?

-Maximum likelihood tree. What samples with 99.9% Native American ancestry were selected after masking? If so, was a missingness filter applied? Of how much and how many sites passed it? The analysis would be more meaningful if run with some migrations.

Figures:

Figure 1: The caption needs to describe better the figure. I think they mistake "branches" by "brackets". A legend that connects colors with language family is needed, interpretation is quite difficult without it. There should be more intuitive way to distinguish in the figure which samples are from Meso and which are from Aridoamerica. With the current way of displaying it is not intuitive at all. % Variance should be shown in each PC.

Figure 2: Again, a better way to intuitively distinguish at first sight between Meso and Arido populations is needed. For panel b) Can populations be sorted in a explicit criteria? It is not clear what the sorting is meant to represent, especially because it is very difficult to read the very tiny population labels. Panel c) "Numbers between branches" -> What branches?. Also, please show the Y axis and specify the value of K.

Figure 3: Panel b) What are colors representing and what are the bars? Longterm -> Longterm. Panel c) Same comment as panel b in Figure2.

Figure 4: Panel a) State what the ML tree is based on (allele frequencies?), show the residual table. Panel b and c) Same comment as Fig.3 panel b. Can they put above plots in c) and d) the topology tested to illustrate what deviations from $D=0$ mean? They do not seem to discuss in the main text why Pame, Huastecos and others have significant (I assume significance but can't be sure given it is not mentioned what the length of the error bars represent, $3de\ 3.3se?$) valued of $D<0$. For panel d) It would be more informative to have the somehow represented what populations constitute the AA and which ones the MA.

María C. Ávila Arcos

Response to reviewers,

We would like to thank you for your helpful suggestions and comments to our manuscript. In response to these thoughtful suggestions, we have revised the manuscript to address all comments, and we believe these have significantly strengthened our manuscript.

The most significant revisions to the manuscript are:

We restructured the manuscript by separating the results from the discussion section, this help us to integrate the results that points to similar conclusions and be less repetitive. We updated the figures with the specifications of the reviewers to be easier to follow through the manuscript, and redone the demographic analysis following the recommendations made by the reviewers. Also, we include a more comprehensive description of the populations and datasets generated in the methods section and include supplementary tables showing the sample size for each analysis.

Second, we addressed the “lack of novelty”, which were suggested by the reviewers #2 and #3, by expanding the discussion of our manuscript, including the comparisons with previous results, and highlighting the new results reported in this work. Also, to explore in more depth the split between populations from the North and South, we compared our Mexican Indigenous populations with ancient genomes from the NNA/ANCB, which helped us to elucidate that the split observed between Aridoamerican and Mesoamerican populations is not due to gene flow from NNA/ANCB to the Aridoamerican populations, confirming our previous claims.

Below please find (in blue) more detailed responses to each original comment of the three reviewers. We hope that the revised manuscript will now be acceptable for publication in *Nature Communications*.

Reviewers' comments:

Reviewer #1 (Remarks to the Author):

García-Ortiz et al. present SNP array data from 706 Indigenous Mexican individuals from 60 out of the 68 recognised ethnic groups in Mexico. By analysing these data jointly with publicly available data from other Indigenous groups in Mexico and the Americas, they explore the genetic structure of Indigenous Mexicans in the context of linguistics and, in particular, geography. The authors find that groups that are closely related in terms of culture and language often form genetic clusters. However, geography appears to be the major driver of genetic structure in Mexico. Based on estimated demographic parameters such as N_e and pairwise population split times, García-Ortiz et al. explore the demographic history of Indigenous Mexicans and conclude that Aridoamerican (AA) and Mesoamerican (MA) populations diverged from each other early on, presumably as a consequence of the adoption of agriculture in MA.

Given the claims presented in this article and especially, the extent of the sample, I consider that this is a valuable work that will be appealing to the broad audience of Nature Communications. However, I would like the authors to include the following suggestions in a revised version.

Thanks for your comments, it was very helpful to strength the manuscript, we next respond to all of them in more detail.

1. I would like the authors to include a more detailed description of the datasets that were used for each analysis. A paragraph or a table summarising the populations (whether they were masked or

not) and number of SNPs per analysis should make it easier for the reader to follow the text. In this regard, please be more specific about what the values in Fig3a and Fig4d represent.

We included a more detailed description of each generated dataset in the methods section starting at line “488” and include supplementary tables 3 and 6 which describe the populations employed in demographic reconstructions and comparison with ancient genomes respectively. Fig. 3a were modified and the sense of the discussion was modified as requested by you and reviewers 2 and 3. Fig4d was removed from this version of the manuscript.

2. Since the manuscript stresses the importance of geography in the determination of the genetic structure of Indigenous Mexicans, Fig1 could be more informative. Please include the the major/main geographic features in Mexico, e.g., mountain ranges. Adding these should make the manuscript more accessible to readers that are not familiar with Mexican geography. Most importantly, highlight AA and MA.

Fig 1 was modified as requested.

3. There are two places in the manuscript where the authors could better discuss the potential effect of gene flow on their inferences.

- A clean split was assumed in order to estimate pairwise population divergence times. Given that estimating the demographic parameters for an isolation with migration model using SNP array data of this kind would be a really demanding task, I would encourage the authors to openly discuss (additional to the statement in l274) this major limitation.

We included the following paragraphs in the discussion section to highlight this limitation:

In line 400: “Furthermore, the assumption of a clean split between populations from Aridoamerica and Mesoamerica can underestimate the observed T due to the relatively recent gene flow between populations from both cultural areas, which may mask an even earlier divergence than what we detected here”

In line 442: “A limitation of the present study is that genetic data were obtained using a commercial microarray that is not designed for Native American populations, which hindered the resolution of the inferences”

- D-statistics of the form $D(\text{Han}, \text{Anzick1}; \text{Karitiana}, \text{Indigenous Mexican})$ are used to claim that AA diverged prior to the Anzick1-MA split. However, this conclusion is not be consistent with the <10ka AA- MA divergence times reported in the paper, in light of the age of Anzick1 (~12.8ka). If Anzick1 forms a clade with MA to the exclusion of AA, then the MA-AA split should predate 12.8ka. Gene flow between AA and a Native American outgroup or between AA and MA excluding the population represented by Anzick1 could be discussed. These two possibilities could be explored through D-statistics of the form $D(\text{Outgroup}, \text{NNA/ANC-B}; \text{AA}, \text{MA})$, $D(\text{Outgroup}, \text{AA}; \text{Anzick1}, \text{MA})$, and $D(\text{Outgroup}, \text{MA}; \text{NNA/ANC-B}, \text{AA})$. If the latter yields non-0 results, it would be interesting to show a plot of $T(\text{AA}, \text{MA})$ as a function of $D(\text{Outgroup}, \text{MA}; \text{NNA/ANC-B}, \text{AA})$ for different (AA,MA) pairs.

In line with this suggestion, we performed these simulations using a Yoruba population as an outgroup and like representatives of NNA/ANC-B ancestry a Lucier and Athabaskan ancient genomes from Scheib et al (ref. 27). Due $D(\text{Outgroup}, \text{NNA/ANC-B}; \text{AA}, \text{MA})$, & $D(\text{Outgroup}, \text{MA}; \text{NNA/ANC-B}, \text{AA})$ solve the same tree topology, we only included the first one in the revised version of the manuscript. Nevertheless, we are attaching the figure from the simulation of $D(\text{Outgroup}, \text{MA}; \text{NNA/ANC-B}, \text{AA})$ to facilitate the revision process.

4. Following the previous comment, it has been shown that some NNA/ANC-B populations have East Asian/Siberian-related ancestry. Therefore, I suggest the usage of an African population in their treemix and D-statistics analyses.

As mentioned in the previous paragraph, we redone TreeMix and D-statistics analyses including the Yoruba population as an outgroup.

5. For the paragraph starting in l222. on Ne, I suggest the authors include a couple of sentences with a disclaimer on how Ne differs from actual census sizes. This could be helpful for the broad readership.

Due the nature of the paper, we believe that the inclusion of this disclaimer is not appropriate.

6. While it is implicit, it is not clear what the authors mean by RoH coefficients. It would be more informative if the authors report more 'standard' RoH metrics such as the proportion of the genome found in RoHs (shown as a boxplot per population).

We agree, we substituted the original supplementary table 4 and instead, we include a boxplot showing the proportion of the genome in ROH in the supplementary figure 8.

In addition, a size analysis of the RoHs per population following Pemberton et al. 2012 (10.1016/j.ajhg.2012.06.014) could be helpful in hypothesising when the isolation took place.

We follow your suggestion and generate the supplementary figure 9, showing the size of ROH in each population. Also, we included the following paragraph in line 232:

“In addition, the categorization of ROH by size showed that all tested Native American populations have a high proportion of short ROH (1-2 Mb), which is consistent with these populations having experienced bottlenecks throughout their history^{22,23}. On the other hand, with the exception of Yaqui, Mazateco from Oaxaca, Chontal from Oaxaca and Maya from Yucatan and Quintana Roo, we observed that all tested populations exhibited different proportions of ROH longer than 8 Mb (Supplementary Fig. 9), which is consistent with the presence of episodes of isolation and/or inbreeding^{22,23}.”

I would have expected a population as isolated as the Tarahumara to carry many long RoHs. Could the authors speculate why this was not the case for the Tarahumara and the other 'divergent populations'?

There are historical records that shows that although Tarahumara can be considered like an isolated population, they have had relationship with several groups from the area, mainly with Guarijio, Pima and in less proportion with Yaqui and Mayo. This contrast with the other divergent populations like Seri and Lacandon in which the geographic barriers of the territories which they inhabited (an island and a jungle respectively) could help to maintain the isolation and then limiting the gene flow with other populations from the area.

7. In l248, the authors talk about the role of geography in the observed genetic patterns. An example would be helpful here, together with the inclusion of geographic features in Fig1.

As previously requested, we modified the Fig. 1 including geographic features. Also, we expanded the discussion concerning the role of geography starting at line 338:

“The geographic pattern was also observed in the admixture analyses in which five of the nine identified components were observed in populations located in northern Mexico, corresponding to

Aridoamerica, which is characterized by a semi-desert climate, whereas the sixth was present in the Lacandon ethnic group located in the Chiapas jungle in southeastern Mexico. Both regions could act as geographic barriers, favoring the isolation of these populations and limiting the gene flow to contribute to the observed genetic structure. This was also observed in the two components detected in the Oto-mange populations, in which the ethnic groups located around the “Eje Neovolcanico” (Fig. 2c, blue component) exhibited similar genetic structure as those located in the state of Oaxaca (Fig. 2c, red component). The ninth was observed in great proportion in populations from the Maya linguistic family. These results support previous hypotheses that geographic barriers, such as semi-desert areas in the north and the mountain ranges, canyons, and jungle regions observed in the Mexican territory, have contributed to the isolation of some populations (Supplementary Fig. 9) by influencing the migration history and the establishment of some populations by limiting gene flow with other groups. Thus, geographic barriers played a major role in shaping the observed patterns of genetic structure in these populations^{2,35,36.}”

8. For the USR1 and Anzick1 genomes, it is not clear how the merging was done. Please include a Methods section with information on whether a genotyping or a random allele were sampling strategy was used. Were any bases trimmed from the ends of the reads?

We included the section “Ancient and modern Native American datasets” in the Methods starting at line 579, which states:

“FastQC files from publicly available ancient genomes from the and present day South Americans were downloaded from their respective repositories^{3,26-29,59}. Adapter sequences, were removed with AdapterRemoval v2.1. The reads were mapped to the human reference genome build 37 using bwa aln v0.6.2-r126^(ref.64) with disabled seeding (-l parameter)⁶⁵. PCR duplicates were removed using picard-tools MarkDuplicates⁶⁶, and local realignment was carried out following the GATK best practices guide^{67,68}. Diploid genotypes were called using HaplotypeCaller from GATK^{67,68}.”

9. The f3-statistics analysis could be presented differently. Given that the intended message is that Indigenous Mexicans are similar to other Native American populations, it would be more informative to compute f3 based on the full worldwide dataset shown in EDFig1. Include figure/table showing that Indigenous Mexicans and other Native Americans have similar values.

We performed the f3-statistics as requested. We included in the Supplementary Fig. 10 and Supplementary Table 7 the comparison with world-wide populations and with Native American populations

10. For the treemix graphs, the placement of Cree and Ojibwa with respect to USR1 and Arctic populations is a bit unexpected. Using CHB as an outgroup could be the cause due to the East Asian/Siberian-related ancestry in some northern populations (see 4.).

As mentioned in 4, we redone TreeMix analyses using a Yoruba Population as outgroup.

The method description for this analysis states that only samples with 99.99% Native American genetic ancestry were used. However, the Cree, Ojibwa and Arctic populations from Reich et al. 2012 have been shown to be heavily admixed. Were these samples treated in a particular way? If not, could Eurasian admixture explain the graphs in EDFig8?

We included 325 samples from Mexico derived from the MAIS cohort which were merged with ancient and modern genomes from the Americas which we termed “Ancient and modern Native American data set”, to clarify the following statement were added at the section methods starting at line 588:

“The gvcf files generated for each ancient genome were called together using the CombineGVCFs command of GATK, converted to plink format using VCFtools^{67,68}, and then converted to eigenstrat format using the convert program from Eigensoft V5.0^(ref.13). Finally, these data were merged with the array data from the 325 genomes from the MAIS cohort with a Native American ancestry >99 (Supplementary Table 6) using the mergeit function from Eigensoft V5.0. Software¹³, yielding a total of 111,586 autosomal SNVs.”

TreeMix analyses were redone with the inclusion of a Yoruba as and outgroup and ancient representatives from NNA/ANC-B ancestry, this help to place USR1 at basal position over the rest of Native American populations. Nevertheless, the Cree still placed above USR1, as you mentioned this may be due the Asian/Siberian-related ancestry in northern populations.

If the 60k site set was used for this analysis, 500 SNP-blocks would result in two few blocks. For f-statistics, ~5Mb-blocks are often used.

In the final estimation we included a 50 SNVs-blocks for TreeMix analyses and 5Mb-blocks for f-statistics. The text specifying these conditions are in the lines 663 & 664 respectively

Finally, did the authors perform multiple optimisations with different starting points? If so, please specify. The same comment goes for the ADMIXTURE analysis.

For TreeMix we do not perform any optimization steep. For admixture we used the default block relaxation algorithm.

To clarify, we included the following text starting at line 599:

“For admixture analysis, we simulated K = 2-16 clusters including cross-validation error estimations and the block relaxation algorithm as the optimization method. For each K, we ran 100 replicates and selected the run with the highest likelihood. Finally, we compared the cross-validation value from each estimation to determine the K that best fit our data”

11. In the local ancestry deconvolution methods section, please specify the masking scheme. Did the authors mask all sites that were not 'homozygous' Native American?

We added the following text at line 561: “Non-homozygous native tracts identified by local ancestry estimation were then masked in each individual by setting the genotypes to missing”

Fig3.

- In D, please use the color scheme in Fig2

Fig 3 was changed as requested

Fig4.

- Please define what the error bars in b,c represent

- Is there a particular reason for using the term 'Outgroup D' - Please use color scheme in Fig2

We defined in Fig 4b and c the error bars including the following statement in the legend “Horizontal bars in (b) and (c) represent 3 standard errors estimated by a weighted block jackknife.”

“Outgroup D” was miss spelling, we changed it for D-statistics

Color scheme in Fig 4 was changed as requested.

I74-75. 'ancient and more recent events' is too vague

The sentence “ancient and more recent events” was removed from the main text

I76. Please describe the number of new genotyped individuals, not the merged dataset.

We modified the text as follows starting at line 76:

“Here, we performed a genome-wide analysis of 716 newly genotyped individuals from 60 of the 68 recognized ethnic groups in Mexico”

I84. 'Mexican/South American indigenous peoples are derived from USR1' is a bit misleading. 'USR1 is an outgroup to' could be easier to follow.

Text was changed as follows: “Upward Sun River 1 (USR1) individual is an outgroup to Mexican/South American Indigenous peoples”

I134. Genetic structure instead of diversity

The referred text was changed as requested and now are located in line 152

I136. Mexican Indigenous populations

The referred text was changed as requested and now are located in line 152

I142. It is not clear that diversity is being 'measured' here. Thus I suggest to lower the tone of this statement.

This text was changed in the revised version of the manuscript as follows: “These results suggest that geographic location influences the genetic structure of these populations” (line 164)

I146. Correspondence instead of correlation (unless there is a correlation score)

This text now is in line 175 and was changed as requested

I165. For clarity, I suggest that the authors highlight the cited examples in Fig2a.

We did not want to make the figure more complex, so we did not highlight these examples

I174. Is there a statistical test for significance here?

We did not perform any statistical test to measure significance, so the text was misleading and was changed for the following expression in line 169 “had the highest levels of genetic differentiation when compared with the other populations based on this statistic”

I216. have shown

The text was moved to the discussion section and was modified in line 369 as follows:

“which also shown a reduction in N_e in recent generations”

I518. Genetic structure instead of variation

The text was changed as requested and now are located in line 597

EDFig9. NortherN branch

ED Fig 9 was moved to Supplementary Fig. 12, figure legend was changed and does not include the description of the NortherN branch

EDFig10. NoRth America

ED Fig10 was moved for convenience, now is supplementary figure 2. The text was changed as requested.

I hope these comments will help the authors improve the manuscript, which I expect to see published in the near future.

Reviewer #2 (Remarks to the Author):

In the paper “Landscape of genomic diversity in Mexican Indigenous populations: insights into the peopling of the Americas” García-Ortiz and colleagues examine the genomic diversity of a large cohort of indigenous Mexican populations (68 groups). Their main finding is a structure between populations from Mesoamerica and Aridoamerica which corresponds to different ecogeographic domains. The populations from Mesoamerica belong to the previously defined ancestry of Amerindian that comprises the rest of South American Native populations, and is close to the ancient remains of Anzick.

The paper is well written and the data released represents possibly the richest survey of Native American diversity for a Latin American country. The north-south structure in Mexico is relevant to understand early migration movements in the continent. Nevertheless, the analysis included in the manuscript are not very conclusive and some of the claims are not sufficiently justified. Most of the results are in fact based on descriptive statistics such as F_{ST} distances and simple calculations of divergence time from N_e and F_{ST} which do not contemplate confidence interval.

The authors should give more insights to read the genetic diversity of this impressive dataset. At the moment it is not clear what element of novelty emerges from the analysis, and the level of descriptiveness and speculation is predominant.

Thanks a lot for your comments, in general based on your suggestions we redone several analyses and made new ones including other Native American populations. This helped us to confirm several of our previous conclusions. Otherwise we expanded the discussion comparing our results with previous works.

To understand the ancestry of the Aridoamerican populations in particular, it would be good to directly compare the data to other ancient data recently published, with a focus on North America, such as Posth et al. 2018 and Scheib et al. 2018.

We compared our data set with individuals from NNA/ ANC-B derived from both papers. The results were discussed in the section “Genetic affinities between modern Native American populations and ancient inhabitants of the Americas” and the results for this comparison help us to discard the possibility of gene flow from NNA/ ANC-B to Aridoamerican populations and thus to confirm our previous claims.

To understand the timing of the north south split with better resolution I suggest to perform demographic simulations with a Bayesian framework such as in Harris et al. 2018.

Although we would have loved to perform a demographic simulation using someone of the Bayesian framework methods, due the nature of our data which was obtained using a commercial genotyped array, the implementation of demographic model becomes a difficult task and the ascertainment bias introduced by the SNP array data could make difficult to obtain credible results. Instead this limitation due the technology used in this work was mentioned in the discussion section at line 442:

“A limitation of the present study is that genetic data were obtained using a commercial microarray that is not designed for Native American populations, which hindered the resolution of the inferences.”

The non-Native American ancestry through the samples is not sufficiently described. This would be important to understand how strong are the claims based on the Native American ancestry alone and then follow which analysis are based on the samples with the high Native American ancestry only.

We included the following text starting at line 141 to clarify the non-Native American ancestry found in our dataset:

“On the other hand, admixture¹⁴ analyses assuming K=4 clusters showed that some Native American samples were admixed with European and African populations, consistent with the history of the Mexican populations. We detected 325 Indigenous samples from the MAIS cohort with at least 0.99 Native American ancestry (Supplementary Fig. 2, upper panel).”

Otherwise, we included this paragraph at line 152 to explain how we deal with the non-Native American ancestry

“Next, to assess the genetic structure of the Mexican Indigenous populations without the recent European and African ancestry, we combined the masked genomes of the Mexican Indigenous individuals from the MAIS cohort with the datasets from Reich et al⁸, Moreno-Estrada et al², and Silva-Zolezzi et al¹², yielding a total of 1,086 individuals.”

Finally, the genetic structure of the country should be better contextualized against what we already know for the continent from previous genetic studies: genomic studies, but also studies based on uniparental markers or HLA types.

We included the following text in the discussion at line 349

“These results support previous hypotheses that geographic barriers, such as semi-desert areas in the north and the mountain ranges, canyons, and jungle regions observed in the Mexican territory, have contributed to the isolation of some populations (Supplementary Fig. 9) by influencing the migration history and the establishment of some populations by limiting gene flow with other groups. Thus, geographic barriers played a major role in shaping the observed patterns of genetic structure in these populations^{2,35,36}.”

In particular, the evidence for a bottleneck around 20 generations ago is not conclusive. From the Supplementary Figure 6, there is evidence of a steady smooth decline for most population, and an increase in size in the last 20 generations. Only populations like LCDN, MXCN, HUI, TNEK, THRA, NAH_MOR show some kind of decline (not that drastic). Many subtle signals of bottleneck point at 25 or 30 generations ago. From this figure alone, it is not possible to point at a strong bottleneck in

the historical time frame of the post-Columbian contact. Demographic testing with simulations could be an option to confirm the validity of a bottleneck in a compatible time frame.

We redone the IBD Ne determination following the pipeline reported by Borrowing et al. In this analysis we phased the populations together, this help to improve the phasing accuracy and as you can see in figure 3 and Supplementary figure 7, there is evidence that the bottleneck is 30 ~18 generations ago both by region and by population tested with a marked decline in the last 20 generations ago. Also, we did this analysis in the raw data set (without masking steep) and in 21 populations with at least 10 individuals with 99 % Native American and the same results were obtained. Although we decided not to include these last two analyses in the last version of the manuscript, we are providing these figures for your reference.

To clarify we modified the methods section in line 631 as follows:

To estimate more recent demographic history, we estimated Ne by identifying IBD tracks based on the pipeline reported by Browning et al²¹. Briefly, the datasets were phased together using Beagle 5.1^(ref.72). IBD segments in the data from each selected population were detected by IBDseq¹⁹. We then used IBDNe¹⁸ for non-parametric estimation of the recent demographic history from IBD segments with the default parameters and a minimum IBD segment length of 2 centiMorgan (cM) to estimate the Ne across generations.

More details comments on the manuscript are following

Abstract: spell out USR1 for non-specialists.

Text was changed as requested.

Introduction:

Expand the introduction by talking about previous knowledge on the genetic structure of Mexico.

We added the following sentence at line 118:

“To the best of our knowledge, few Mexican ethnic groups have been examined at a whole genome level, yet a complex genetic structure has been observed in those few groups^{2,5,8}”

Line 115: why the most diverse population in America is in Mexico?

We changed the text to slow the tone of our asseveration in line 114:

“Despite being one of the largest and diverse groups in America”

Results:

Mention how many SNPs are included in the analysis after merging the data. In general, mention if the whole set of populations is included in each analysis, or which part of it.

We expand the methods section specifying how many samples and SNVs were included in each analysis and in the main text we added

In line 135: “First, we compared our 716 Mexican Indigenous individuals to worldwide populations, including Mexican Native American populations previously reported by Reich et al⁸, Moreno-Estrada

et al², and Silva-Zolezzi et al¹². The merged dataset comprised 3,490 individuals from 218 populations and 61,393 autosomal SNVs. Principal component analysis (PCA)¹³”

In line 154 “Indigenous individuals from the MAIS cohort with the datasets from Reich et al⁸, Moreno-Estrada et al², and Silva-Zolezzi et al¹², yielding a total of 1,086 individuals.”

In line 205: “To track the demographic histories of Indigenous Mexican populations, we estimated the effective population size (Ne) across time based on two different methods in 48 ethnic groups from the masked datasets with sample sizes of at least 10 individuals genotyped with the Affymetrix Genome-Wide Human SNP Array 6.0 (Supplementary Table 3).”

In line 266: “We then combined the 325 indigenous samples with seven ancient genomes from America (Supplementary Table 6), yielding a total of 111,586 autosomal SNVs.”

Line 141: which hypothesis? From which evidence?

The text was moved to the discussion section and now starts at line 349 and was modified as follows:

“These results support previous hypotheses that geographic barriers, such as semi-desert areas in the north and the mountain ranges, canyons, and jungle regions observed in the Mexican territory, have contributed to the isolation of some populations (Supplementary Fig. 9) by influencing the migration history and the establishment of some populations by limiting gene flow with other groups. Thus, geographic barriers played a major role in shaping the observed patterns of genetic structure in these populations^{2,35,36}.”

142: why are we talking about geography in shaping diversity? We only see three blocks corresponding to three major regions, but the geographic differences of this structure are not explained.

We integrated the results of PCA, NJ and Admixture to get our conclusion in the discussion section as mentioned above (line 349)

The FST based NJ tree is too simplistic to draw major conclusions based on the linguistic affiliation of the samples. The methods say that bootstrap was performed: where can we see this? How much is this “high clustering” of linguistic families? From the tree in particular it is impossible to see if there was any linguistic radiation, and reflect on the demographic spread hypothesis of the Uto-Atztecans suggested by Diamond and Bellwood. More linguistic reference and hypothesis should be introduced to make proper claims on linguistic-genetic relationships. It should be checked if in some cases populations who speak a language of the same family cluster together despite being geographically distant, or if a population is closer to a distant one (but linguistically related) than to a non-linguistically related neighbor.

To answer your questions: first we slow the tone of the claims we were done about linguistic and genetic correlations calling them just “correspondence” (line 175)

Second, we modified the Fig 2b adding the bootstrap values to the NJ tree. Also, the methods section was modified to describe the process we used to generate the bootstrap replicates in a better way with the following text starting at line 609: “To generate bootstrap replicates of the NJ tree, we randomly removed 100 times 1% of the total SNVs of the masked dataset. With each set of remaining variants, we generated 100 independent pairwise-FST matrices to generate a new NJ tree from each one (replicates) in R package APE V5.1. The consensus tree was generated under DendroPy V 4.0.0(ref.70) using the whole data NJ tree to maintain the topology, and each replicate was used to obtain the bootstrap values.”

Regarding on Diamond and Bellwood. You are right, it is not possible to see the radial expansion suggested by Bellwood, then we remove that sentence from the main text.

We included the following text as example of cultural/linguistic in line 331: "Otherwise, the pattern observed in the NJ tree, in which individuals from related languages cluster together (Fig. 2a), is compatible with previous anthropological studies showing that human cultural practices, such as language and human cooperation, as well as many other cultural features, have modified environmental conditions, triggering changes in allele frequencies and contributing to differentiation of the populations²⁹⁻³⁴."

We do not observe that phenomena in which linguistically related but distant ones fall together in the tree. Just for Popoluca de la Sierra and Zoque but they are relatively close each other.

It would be good to discuss a bit more the groups coming from ADMIXTURE analysis. is there any cultural, regional or historical region for these clusters?

We expand the discussion about the admixture analyses starting at line 338:

"The geographic pattern was also observed in the admixture analyses in which five of the nine identified components were observed in populations located in northern Mexico, corresponding to Aridoamerica, which is characterized by a semi-desert climate, whereas the sixth was present in the Lacandon ethnic group located in the Chiapas jungle in southeastern Mexico. Both regions could act as geographic barriers, favoring the isolation of these populations and limiting the gene flow to contribute to the observed genetic structure. This was also observed in the two components detected in the Oto-mange populations, in which the ethnic groups located around the "Eje Neovolcanico" (Fig. 2c, blue component) exhibited similar genetic structure as those located in the state of Oaxaca (Fig. 2c, red component). The ninth was observed in great proportion in populations from the Maya linguistic family. These results support previous hypotheses that geographic barriers, such as semi-desert areas in the north and the mountain ranges, canyons, and jungle regions observed in the Mexican territory, have contributed to the isolation of some populations (Supplementary Fig. 9) by influencing the migration history and the establishment of some populations by limiting gene flow with other groups. Thus, geographic barriers played a major role in shaping the observed patterns of genetic structure in these populations^{2,35,36}."

Line 196-199: the difference between the second and third hypothesis is not clear. They both concern relatively recent gene-flow.

Yes, both hypotheses are highly related, we modified the text as follows to unify them in line 358:

"In addition, these correlations can reflect structured patterns of interaction and genetic exchange, leading to gene flow between groups in proximity with one another, possibly reflecting the influence of alliances and migrations that occurred during different periods in the history of Mesoamerica³⁷⁻³⁹"

The difference between the two methods to detect N_e should be better explained. Which method was used to calculate long-term N_e ? As said before, the descriptive interpretation of these plot is not corresponding to the apparently clear patter described. Definitely not with a reduction of 90% of the population as referred to historical records.

We modified the results section explaining which methods were used for each determination in line 205 we added the following

"To track the demographic histories of Indigenous Mexican populations, we estimated the effective population size (N_e) across time based on two different methods in 48 ethnic groups with sample sizes of at least 10 individuals (Supplementary Table 3). Demographic reconstructions based on

linkage disequilibrium (LD) analysis^{16,17} showed little evidence of a fluctuation in N_e before 150 generations ago (Supplementary Fig. 6). To evaluate more recent demographic changes, we estimated the N_e based on identity by descent (IBD) tracks implemented in the software IBDNe^{18,19}

In line 217 we clarify how we estimate long-term N_e :

“Next, we estimated the long-term N_e based on LD patterns using Neon Software^{16,17}.”

Line 218-220: not clear which changes in allele frequencies and how could be linked to current genetic diversity.

This text was removed from the revised version of the manuscript

Line 222: how was this result obtained?

As mentioned above, we added the following text to clarify in line 217: “Next, we estimated the long-term N_e based on LD patterns using Neon Software^{16,17}.”

About the ROH analysis: what are the numbers in Supplementary Table 4? Is it the length? Is it proportion? How long are these long ROHs? This result should be displayed with a clearer output. See for example Schroeder et al. 2018 (PNAS). At the moment it is not clear what it adds to the results.

The ROH values in Supplementary Table 4 are the proportion of the genome in ROH. We removed this table from the revised version of the manuscript and adopt a more standardized approach to evaluate ROH. First, we show the proportion of the genome in ROH as a boxplot (supplementary figure 8) and generate the supplementary figure 9, showing the size of ROH in each population. Also, we included the following paragraph in line 232:

“In addition, the categorization of ROH by size showed that all tested Native American populations have a high proportion of short ROH (1-2 Mb), which is consistent with these populations having experienced bottlenecks throughout their history^{22,23}. On the other hand, with the exception of Yaqui, Mazateco from Oaxaca, Chontal from Oaxaca and Maya from Yucatan and Quintana Roo, we observed that all tested populations exhibited different proportions of ROH longer than 8 Mb (Supplementary Fig. 9), which is consistent with the presence of episodes of isolation and/or inbreeding^{22,23}.”

Divergence time:

Mention how high are the highest T . is it compatible with the history of the region?

In line 245 we added this sentence: “the uppermost value of T was observed between Seri and Maya from Quintana Roo ($T = 11.8$ Ka, Supplementary Table 5).”

This is consistent with the differentiation of populations from the North and south of Mexico

Line 248: which mountains? Where are those? Better explain about potential geographic barriers. Would it be relevant to show them on a map (like fig.1)? is the distribution of pairwise divergent time coherent between the grouping proposed in Fig 3d? it is difficult to see patterns from Fig 3c.

Starting at line 349 we added the following text about geographic barriers: “These results support previous hypotheses that geographic barriers, such as semi-desert areas in the north and the mountain ranges, canyons, and jungle regions observed in the Mexican territory, have contributed

to the isolation of some populations (Supplementary Fig. 9) by influencing the migration history and the establishment of some populations by limiting gene flow with other groups. Thus, geographic barriers played a major role in shaping the observed patterns of genetic structure in these populations^{2,35,36.}”

Also, we modified Fig 1 adding the geographic characteristics to the map. Finally, we ordered Fig 3c to match with the north to south pattern and with Fig 4d.

Because the method relies solely on FST distances and Ne and it is not adjusted for uncertainty and pairwise migration, and no confidence interval is reported, it would be important to see how robust is the pattern overall and how it could be driven by drift and contact. For example, it is not clear how the values in line 264 are calculated.

In line 264 from the original paper we were reporting the arithmetic mean of the observed divergence time. In the revised version of the paper in line 248 we are reporting the range as follows: “Populations from northern Mexico corresponding to Aridoamerica diverged from the populations in the center/south around 3.96 to 9.47 Ka ago and from the populations in the southeast approximately 4.84 to 10.15 Ka ago (Fig. 3c and 3d, Supplementary Table 5).”

We are aware of the limitation of this approach, based on this we added the following text to the revised manuscript at line 400: “Furthermore, the assumption of a clean split between populations from Aridoamerica and Mesoamerica can underestimate the observed T due to the relatively recent gene flow between populations from both cultural areas, which may mask an even earlier divergence than what we detected here.”

The divergence dates should be discussed against other results for the Americas, such as Harris et al. 2018, for example.

We compared our results with the reported by Avila-Arcos, M. C. et al (ref 22) which show similars T

Line 282: be specific about these streams and the previous literature that proposed them.

We added the references 3,8,53,54 to this text at line 407.

Line 294-295: this statement is particularly vague.

This statement was slightly modified in line 418, to give us entrance to the discussion on the comparison of modern and ancient Native Americans

“suggest that the history and ancestry of Indigenous populations in Mexico may be more complex than reported”

Line 297: mention how many SNPs are left after merging with ancient DNA data.

Text was modified at line 266 as follows:

“We then combined the 325 indigenous samples with seven ancient genomes from America and South American populations^{3,28} (Supplementary Table 6), yielding a total of 111,586 autosomal SNVs.”

Line 319-324: this statement is particularly vague.

This text was removed from the final version of the manuscript

Discussion:

In general, the discussion is not adding much perspective to the previously discussed results.

We modified the discussion. First, we removed from the results the sections that sound better for the discussion, this helped to expand the discussion and integrate the discussion in better way.

Line 340: how is the complex geography contributing to population diversity? In which sense is the geography of Mexico complex?

This text was removed from the revised version of the manuscript. Instead, as mentioned previously, we added a discussion about the geographic barriers in Mexico (lines 349 – 355).

Line 350: the link is not clear, it seems very speculative.

This text was removed from the revised version of the manuscript

Line 360-363: provide references and explain this statement.

This argument was resumed in line 396 – 400 in the following sense:

“Notably, other recent studies have provided evidence that trade, political relationships, and local sociocultural histories have shaped the demographic histories and migration patterns, mainly in the classic and post-classic period in northern, western, and central Mexico, influencing gene flow patterns among populations and the population structures^{37,50-52}”

And the reference 37 & 50 -52 were used to justify these claims.

The last part of the discussion is a repetition of concepts already expressed elsewhere in the manuscript and broad speculations.

As we mentioned, we modified the discussion in a more integrative and less repetitive fashion

Methods:

The use of linguistic labels should be better justified. For example, the Yuto-Nahua is commonly referred as Uto-Atzecan, while the Oto-Mange is referred as Oto-Manguéan.

We try to explain this in the introduction at line 105:

“Today, 68 recognized ethnic groups are clustered into 11 linguistic families⁷”

And in the results at line 176:

showed a correspondence between geographic distribution and linguistic classification proposed by the National Institute of Indigenous Languages in Mexico (INALI; from the Spanish Instituto Nacional de las Lenguas Indígenas)⁷

Explain which samples are newly published and separate them from published data. Mention which Native American published data was involved, and which data from other continents and why. For example, it is not clear which reference was used for non-Native American admixture.

We clarify this in the abstract starting at line 76 :” Here, we performed a genome-wide analysis of 716 newly genotyped individuals from 60 of the 68 recognized ethnic groups in Mexico.”

In line 135: “First, we compared our 716 Mexican Indigenous individuals to worldwide populations, including Mexican Native American populations previously reported by Reich et al⁸, Moreno-Estrada et al², and Silva-Zolezzi et al¹².”

In line 497: “From this cohort, a total of 716 unrelated individuals belonging to 71 Indigenous communities representing 60 ethnic groups from 10 linguistic families were selected for genome-wide genotyping based on the availability of samples”

In line 522: “First, we merged our 716 Mexican Indigenous samples with world-wide populations reported by Reich et al⁸, Moreno-Estrada et al², and Silva-Zolezzi et al¹². After QC in each dataset, all data were merged using the mergeit function of the Eigensoft V5.0. Software¹³”

The order of the methods does not follow the order of the results. The global ancestry estimation should be discussed first in the results, as well.

We added the results about the global ancestry in the results and modified the methods section to match with the order of the results.

What is the difference between the analysis mentioned in line 502 and the one in line 514?

In line 502 we performed the admixture analysis in the dataset without masking step. In line 514 we performed the admixture analysis in the masked dataset.

To clarify, we modified the methods section to better explanation starting at line 521, section “**Mexican Natives modern populations dataset**” and in the section “**Mexican Indigenous masked datasets**” starting at line 547.

Which N_e was used to calculate divergence time?

Long-term N_e was used to this calculation, in line 241 we state that:

“Both the long-term N_e and F_{ST} between pairs of populations were employed to calculate the divergence time between populations in generations (T)”

In the methods section in line 644:

“The long-term N_e was estimated based on LD patterns using the method reported by McEvoy et al¹⁶ in R package NeON¹⁷.”

Is the Treemix analysis useful at all? It is mentioned solely to confirm a basal position for USR1. Supplementary figure 9 with migrations is not really discussed in the text.

You are right, we were not using at all these results, we add a discussion about this in line 431, now Supplementary figure 9 is Supplementary figure 12.

“A TreeMix admixture graph allowing 20 migration edges inferred multiple waves of gene flow from Mesoamerican to Aridoamerican populations and vice versa, mainly from Mesoamerican populations to the branch related to Cucapa and Huichol from Aridoamerica (Supplementary Fig. 12), which is consistent with the first scenario.”

Main figures:

Fig. 2: which order are the populations in the heatplot? Can the order be the same of the populations in Fig1c Legend? So it is easier to track down similarities.

Fig. 2 was changed as requested.

What is the color coding corresponding to in Fig. 3b, 4b and 4c? can the order of the populations be consistent?

Fig. 3b, 4b and 4c was changed as requested maintaining the same color and order showed in Fig 1.

Reviewer #3 (Remarks to the Author):

The manuscript “Landscape of genomic diversity in Mexican Indigenous populations: insights into the peopling of the Americas” presents population genetics analyses based on genotyping array data from newly-generated as well as previously published data belonging to Indigenous populations from Mexico. Of value, the authors have made an important contribution by expanding the catalogue of Mexican genetic variation to more linguistic groups. The results from the analyses carried out in this study, have few new implications regarding the peopling of the Americas. Although the work has some limitations, the study has potential as it provides newly testable hypotheses that can be addressed with less ascertained (i.e. non array) data is available. As it stands, I feel the work lacks enough information to make it reproducible, therefore I have several minor and few major comments about the work that, if addressed I believe could improve the manuscript and be useful to the scientific community interested in the field.

Dear Dr. María Ávila Arcos, thanks for your comments. We believe that we have responded to all of your comments appropriately. Please in the next lines find the specific responses to each one.

General comments:

It would be more informative to have a better definition of what is Meso and What is Aridoamerica, the current description of the limits by saying “Center and South, and North” is very ambiguous, perhaps highlighting the regions in the map from Figure 1 would help. Also the authors want to mention that these regions have shifted as a result of climatic fluctuations in the last 1000 years.

We expanded the explanation about the two regions (line 107 – 113)

“These populations can be divided into two main geographic/cultural areas: Mesoamerica and Aridoamerica. Mesoamerica comprises central and southern Mexico, and during the pre-Hispanic era was inhabited by sedentary agricultural societies favored by the great biodiversity of this region. Aridoamerica encompasses a semiarid area in northern Mexico that preserved nomadic forms of subsistence throughout the pre-Hispanic era.”

Also, we modified Fig 1 adding the Aridoamerica/Mesoamerica division to the map.

I believe the authors should have a statement of the limitations of using genotype and the risk of ascertainment bias, given the arrays used were not designed with Latin American populations (let alone Native Indigenous populations) in mind. They should mention what they did to control for this in each analysis or justify why this should not be concern. Along these lines, the authors mention in several occasions the implications of their results (and previous) in light of accumulation of

population-specific private alleles. However they do not acknowledge that their approach is unable to capture this.

Certainly, we are aware that the genotyping method is the main limitation of this work. We discussed the limitation of this approach at line 442:

“A limitation of the present study is that genetic data were obtained using a commercial microarray that is not designed for Native American populations, which hindered the resolution of the inferences.”

Sometimes arguments that are more suitable for the discussion are introduced in the results, making it confusing to distinguish what is an actual results from the study and what is information just needed for discussion of results (e.g. discussion in lines 255-260 that are found in results). This is common throughout the text so I encourage the authors to revisit the manuscript and delimit better the sections.

As mentioned at the beginning of this letter, we restructured the manuscript separating the results from the discussion.

Abstract:

“than has been recognized so far” -> “than what has been recognized so far” (though I recommend a better phrasing.

The text was changed as follows in line 87: “than recognized thus far”

Introduction:

-The authors make the claim throughout the manuscript that this is the largest cohort of Indigenous individuals sampled to date. I believe it is important from the introduction to clarify unambiguously what is the actual contribution of new data, as opposed to previously published datasets. In line 124 they state:

“Here we report the results of a population genetic study on the largest cohort of Indigenous individuals sampled in Mexico to date. We included a total of 1,086 individuals from 60 different Mexican ethnic groups, representing all linguistic families [...] Genotypes from Mexican Indigenous populations were compared with those from 146 previously published worldwide populations^{2,8,9}.”

The sentence can be misleading as it could imply that the authors have generated data from 1,086 Indigenous individuals themselves. I believe that for sake of recognizing previous work it should be stated unambiguously and from the very beginning, (i.e. here) that the data that was generated for this study comprises 633 individuals, while the remaining is from previous studies. This information is only described in the Methods section, but it should be clear from the introduction.

To a better understanding on what samples are new, we modify the redaction of that sentence as follows:

“We genotyped at the whole genome level 716 individuals from 60 of the 68 recognized ethnic groups in Mexico belonging to the MAIS cohort⁹⁻¹¹, which were merged with previously published datasets, yielding a total of 1086 Native Americans from Mexico, representing all linguistic families except Kickapoo (Algonquian language family) (Fig. 1a, Supplementary Table 1).”

Results

-Line 153: don't -> do not

This text was removed from the reviewed version of the manuscript

-It is not clear what is meant from the statement:

“While Bellwood proposed that speakers of the initial Yuto-nahua and Mayan languages spread far outward from central Mexico once they had become farmers, it is not clear that there is strong evidence for the linguistic radiations he proposed”

Specifically “it is not clear that there is strong evidence”, does the authors mean it is not clear from their data? If so, is it correct to say that because a genetic signal is not observed, then a linguistic radiation likely did not take place? I would argue that language can be diffused culturally, without a strict need of gene flow, therefore I would suggest the authors to review this sentence.

This text was removed by the revised version of the manuscript

-Lines 160 – 161: “They do not do so perfectly” -> It is unclear what is meant by this. What would a “perfect” clustering be?

This text was removed by the revised version of the manuscript

-Line 172: “Pairwise F_{ST} comparisons identified the Tarahumara, Guarijio, and Cucapah in the North of Mexico, and those previously published like the Seri (North) and Lacandon (Southeast)² as significantly different genetically (Fig. 2b).”

It is unclear from the phrasing what are the “pairs” are in the “Pairwise F_{ST} comparisons”.

We are not comparing specific pair of populations, we are trying to express that these populations exhibit highest pairwise F_{ST} Values when they are compared with the other tested populations. To clarify, we modified the text as follows in line 167:

“Otherwise, pairwise F_{ST} comparisons identified the Tarahumara, Pima, Guarijio, and Cucapa in northern Mexico, and previously published populations, such as the Seri (North) and Lacandon (Southeast)², had the highest levels of genetic differentiation when compared with the other populations based on this statistic”

-Lines 183 – 186: “Based on this K, six of these clusters were mainly observed in a single population (Seri, Tarahumara, Pima, Tepehuano, Huichol and Lacandon), while two were mainly observed in several ethnic groups inhabiting the Centre/South (here referred as multi-ethnics), and one in populations in the Southeast that are part of the Mayan linguistic family (Fig. 2c)”.

As is, the sentence implies that six clusters were observed in a single population. i.e. that single population has six clusters. I only managed to understand the sentence when looking at the Figure. I believe the sentence can be rewritten to better convey the message of Figure 2c.

We rewrite this sentence as follows in line 192 -194:

“Based on this K, six of these clusters were mainly limited to a single population (Seri, Tarahumara, Pima, Tepehuano, Huichol, and Lacandon).”

-Line 191: “These data, along with the Mantel test, strongly support the notion that geography has played a crucial role in shaping the genetic diversity across these populations”

Could authors add that this has already been suggested by previous genetic studies?

This sentence was removed from results section. Instead, we integrated the results from PCA, admixture and mantel test in the discussion. The conclusion of this part states the follows at line 349 -355

“These results support previous hypotheses that geographic barriers, such as semi-desert areas in the north and the mountain ranges, canyons, and jungle regions observed in the Mexican territory, have contributed to the isolation of some populations (Supplementary Fig. 9) by influencing the migration history and the establishment of some populations by limiting gene flow with other groups. Thus, geographic barriers played a major role in shaping the observed patterns of genetic structure in these populations^{2,35,36}.”

-In the paragraph comprising lines 191-199 the authors propose three alternatives to explain the genetic-geographic correlations. They say these:

- 1) may reflect ancestral relationships and initial settlement patterns, where closely related people settled in the same geographic areas, and their descendants persist in those regions until today.
- 2) can reflect structured patterns of interaction and genetic exchange, leading to gene flow between groups in proximity with one another.
- 3) could also reflect the influence of alliances and migrations that took place during different periods in the history of Mesoamerica.

I actually think all three are related, population continuity of people settled in the same geographic area implies gene flows between groups in proximity with one another. As to the third alternative, they mention migrations, which if long distance would then not reflect the genetic-geographic correlations, unless they mean migrations between groups in proximity, which falls back to points 1 and 2. I think authors should be clearer in discussing this point.

We unified the text starting at line 358 this context as follows: “In addition, these correlations can reflect structured patterns of interaction and genetic exchange, leading to gene flow between groups in proximity with one another, possibly reflecting the influence of alliances and migrations that occurred during different periods in the history of Mesoamerica³⁷⁻³⁹”

-“Based on historical records that provide insights into the demographic processes in specific regions, some authors have suggested that the total Native American population size in that time decreased by more than 90% at this time. In addition, recent studies based on ancient and modern Native American genomes have showed a significant reduction of ~50% in the Ne. Our results are consistent with these previous observations.”

In light of previous reports, is it possible to add a % estimate of Ne reduction in the data here analyzed?

Although we really like to add a value for the estimated reduction, we believe that it is not possible to asses with the methodology employed.

-Can the authors lay succinctly the basic underlying principles of the IBDNe approach?

We added an explanation in two sections of the revised manuscript:

In the results at line 211: “To evaluate more recent demographic changes, we estimated the N_e based on identity by descent (IBD) tracks implemented in the software IBDNe^{18,19}.”

In the methods in line 631: “To estimate more recent demographic history, we estimated N_e by identifying IBD tracks based on the pipeline reported by Browning et al²¹. Briefly, the datasets were phased together using Beagle 5.1^(ref.72). IBD segments in the data from each selected population were detected by IBDseq¹⁹. We then used IBDNe¹⁸ for non-parametric estimation of the recent demographic history from IBD segments with the default parameters and a minimum IBD segment length of 2 centiMorgan (cM) to estimate the N_e across generations.”

-Line 222: What is meant by “mean long-term N_e ”?

It was misspelling, “mean” was deleted from the text at line 218.

-When discussing population-specific N_e , the authors might want to consider a recent previous report (<https://www.biorxiv.org/content/biorxiv/early/2019/01/30/534818.full.pdf>) that calculated N_e in Huichol, Tarahumara, Triqui and Rarámuri (Tarahumara) and contrast to this.

We added the following text at line 223: “and are similar for previously studied populations such like Tarahumara, Huichol, Triqui and Mayas⁴⁹”.

-Line 239: “The most divergent populations have the highest value for T (Seri, Guarijio, Tarahumara, and Lacandon; Fig. 3d, supplementary Table 5)”

It is not clear in respect to which other populations have the highest values.

Like with the F_{ST} , we are not referring to specific pairs of populations, instead we are referring that this populations shows the highest values of T whit the other populations tested.

-Line 264: The divergence times are interesting. Since the authors have already calculated it, could they state estimated divergence estimates between Central/South and Southeast. Also, they might also want to consider and discuss in the context of previous estimations done in <https://www.biorxiv.org/content/biorxiv/early/2019/01/30/534818.full.pdf>, which are concordant and further support, their results.

We added the following text at the discussion section starting at line 387: “These results agree with a recent study of 76 masked exomes from Native American populations from Mexico that found a similar T between populations from northern and central/southern Mexico⁴⁹”

-“Furthermore, recent evidence based on ancient DNA studies also reveals a much more complex history of migrations, divergence, and large-scale movements between North and South America, supporting our findings from the study of present-day Native population in Mexico.”

Can authors be more specific about the “complex events” that support their findings?

This text was removed from the main text.

Discussion

-“Our results show that the genomic variation observed in Mexican Indigenous populations is the result of complex interactions between geographic, cultural, and demographic events that have shaped the genetic architecture of these populations.”

In fairness, genomic variation of all human populations is the “result of complex interactions between geographic, cultural, and demographic events”. Their results show interesting patterns regarding split times and N_e fluctuations, but really as they don’t directly assess (but rather speculate—sometimes righteously) about the role of past events, I believe the sentence above is fat stretched, even more considering that previous reports on genomic data from Indigenous individuals have already shown many of the patterns here revealed.

We agree with you, but considered that this is the first time that the vast majority of Native American Populations are studied, we believe it is worthy to note that the same trend is observed in all the populations studied and are not restricted to the few groups studied in the previously valuable research.

-“The complex geography of the Mexican territory seems to have contributed to the isolation of some groups, such as Seris and Lacandons, by limiting gene flow and favouring the enrichment of local private alleles that are rare or absent elsewhere.”

This is consistent with theory, and has been proposed already elsewhere (Moreno-Estrada 2014), but has not been really shown as part of this study as it is not possible to address with genotype data.

This text was removed from the revised version of the paper.

- “we observed that all tested populations experienced a strong bottleneck around 20 generations ago, consistent with the beginning of the European colonization and conquests that disrupted native societies and led to the demographic decline of these populations”

I think this is a very neat, although anticipated, observation. I believe it merits some discussion in the context of the paper by Browning et al. <https://journals.plos.org/plosgenetics/article?id=10.1371/journal.pgen.1007385>

In which they do not find such a bottleneck when performing an ancestry-specific N_e estimation through time in admixed Mexican population, but they do in other American populations. Along these lines I think it is also necessary to discuss these observations and the argument made several times about the accumulation of private (potentially deleterious) alleles in the context of present-day admixed Mexican populations. In light of this, it would be very interesting also to observe the difference between N_e calculation through time in individuals that are >99.9% indigenous, and those with more admixture with and without a masking step.

We included in the discussion the following paragraph at line 363 including the Browning’s paper.

“In this case, we observed a decline in the N_e of all tested populations between 15 - 30 generations ago (Fig. 3a, Supplementary Fig. 7), followed by an expansion. Although our results contrast studies investigating the population history of Native Americans through admixed populations from America²¹, our analyses are in agreement with recent studies modeling demographic reconstructions through time based on mitochondrial^{20,40} and whole exome data from ancient and modern Native American samples^{20,40}, which also show a reduction in N_e in recent generations. In most cases, the timing of the observed bottlenecks corresponds with the beginning of the European colonization of the Americas (Supplementary Fig. 7) and is consistent with prior studies on the impact of settler colonialism on Indigenous communities^{20,40} and historical records that provide insights into the demographic processes in specific regions⁴¹⁻⁴³. Some authors have suggested that the total Native American population decreased by more than 90% at this time⁴¹⁻⁴³. Overall, our

findings show for the first time that the strength and timing of the contraction observed in Indigenous populations was not localized to a particular population but, instead, has been widespread with a different impact in the tested populations, reflecting the different demographic histories in each population.”

Regarding on the calculation of N_e on the unmasked and the 99 % datasets, we observed the same bottleneck. Although we decided to not include this last two simulation in the final version of the manuscript, we are attaching the figures at the end of this document for your revision.

-Lines 361-363: In addition, previous studies have pointed to the complex relationships ...” Can they cite those previous studies?

This text was moved to the discussion at line 396 - 400:

“Notably, other recent studies have provided evidence that trade, political relationships, and local sociocultural histories have shaped the demographic histories and migration patterns, mainly in the classic and post-classic period in northern, western, and central Mexico, influencing gene flow patterns among populations and the population structures^{37,50-52”}

And the reference 37 & 50 to 52 were used to justify these claims.

-Line 365: “the comparison of modern Indigenous populations from Mexico with the ancient genomes from USR1 and Anzick-1 suggests that populations from Mexico came from the same genetic stream as the First Peoples to inhabit the Americas 8,45, but that these populations experienced an early split between the Aridoamerica and Mesoamerica/South America branches in the settlement of North America (Fig. 4).”

I think this is perhaps the most important and novel contribution of the paper, therefore I think it needs further testing and better quality discussion.

We expand the discussion about this at line 421 as follows:

“In this sense, the comparison of modern Indigenous populations from Mexico to the ancient genomes from USR1, Lucier, Athabaskan, and Anzick-1 show that Native Americans were derived from a common ancestor related to a population closer to USR1, consistent with the “First American” dispersal model^{8,26}, and that the Mexican Indigenous populations experienced one of the following scenarios: 1) a split between Aridoamerica and Mesoamerica occurred prior to the split between Mesoamerica and Anzick-1 with posterior gene flow between the Aridoamerican and Mesoamerican populations, or 2) a split between Aridoamerica and Mesoamerica occurred after they split from the Anzick, with Aridoamerican populations carrying gene flow from a population that split above the population represented by Anzick-1 and below the NNA/ANC-B branch (Fig. 4, Supplementary Fig. 16). A TreeMix admixture graph allowing 20 migration edges inferred multiple waves of gene flow from Mesoamerican to Aridoamerican populations and vice versa, mainly from Mesoamerican populations to the branch related to Cucapa and Huichol from Aridoamerica (Supplementary Fig. 12), which is consistent with the first scenario.”

Because the data is derived from different platforms (sequencing in the ancient, versus genotyping in modern), and since it has been shown that comparing platforms/methods could introduce some bias (as more similarity between two samples/populations could be product of these being processed on the same platform). Is Karitiana sequenced data or genotyping? Authors should describe at length why they do not think this is affecting their analyses and prove it. They should also discuss how ascertainment in the genotyping arrays could be affecting their results. Furthermore, the test is made with a Han sample, I believe that a better outgroup should be a Sub-saharan African. Actually from figure 4 it is not crystal clear that all AA populations are closer to

USR1 than to Anzick1. Maybe is only because of a problem representing the results in the figure (see comments specific to this figure below).

We redone the D-statistics analyses following your and the other reviewer's recommendations using a Yoruba population as outgroup. To deal with the concern of the bias introduced by combining SNP array & sequencing data, we performed several simulations including the following population: The original data from Karitiana presented by Reich et al⁸, and two Karitiana and three Quechua sequenced from Mallick et al 2016 and two sequenced Aymara from Raghavan et al³. Also, as internal control we decided to use the Tzotzil population from our dataset. In all the simulations in which the representative from South-American populations was either, Karitiana genotyped, Karitiana and Aymara sequenced and Tzotzil, the same trend was observed. Due to these results we decided to use the Karitiana and Aymara sequenced and the Tzotzil from our dataset to demonstrate the proposed topology in the D-statistics (Fig 4c & d and supplementary Fig 16). To help with the revision process, we are attaching the D-statistics simulation using the genotyped Karitiana⁸ as a South-American representative.

##Methods

-For each analysis please be explicit about the number of samples used (ideally referencing a table with the individuals per analysis), the exact filters, and the number of sites passing.

We added the sample size and SNVs number to each section.

In addition to supplementary table 1, we generated supplementary table 3 and 6 in which we are specifying the exact number of samples per population employed in each analysis.

-The methods section needs major improvements, many details needed for reproducibility are lacking:

In general, we rewrite the methods section to be more specific about the samples and methodology employed to do our estimations.

-Mention which local and research committees (IRBs?) approved the protocol and report the number of the project approved. This is particularly important given the work deals with very vulnerable and marginalized populations, so many ethical considerations are in place and need to have been approved by IRBs and have been consulted with local Indigenous groups.

The following paragraph was added at line 504-510: "This study was designed in accordance with the Declaration of Helsinki and approved by the Research, Ethics, and Biosafety Human Committees of the Instituto Nacional de Medicina Genómica (INMEGEN) in Mexico City (protocol number 31/2011/I) with the support of the National Commission for the Development of Indigenous Communities (CDI, from the Spanish "Comisión Nacional para el Desarrollo de Pueblos Indígenas") and with the agreement of the Indigenous leaders from each community."

-Line 478: "A subsample of MAIS" → subsample of what size

We clarify this text in line 497 as follows: "From this cohort, a total of 716 unrelated individuals belonging to 71 Indigenous communities representing 60 ethnic groups from 10 linguistic families were selected for genome-wide genotyping based on the availability of samples (when possible we selected at least 10 members per ethnic group, Supplementary Table 1). From this group, 644 samples were genotyped using the Affymetrix Human 6.0 array, and 72 samples were genotyped using the Illumina OMNI 2.5 array."

-Line 492: “ $p < 0.001$ ” -> what test?

The $p < 0.001$ is a standard cut/off used in genome wide association studies to avoid bias due the different source/genotyping procedure, the test used is a chi-square.

-More details are needed for ADMIXTURE, where there any “pruning” steps? Did they run it only once? Admixture needs to be run several times (recommended between 10 and 100) and then runs need to be condensed or the one with the best Log Likelihood be selected. There is no mention whatsoever of this.

We added this description starting at line 599: “For admixture analysis, we simulated $K = 2-16$ clusters including cross-validation error estimations and the block relaxation algorithm as the optimization method. For each K , we ran 100 replicates and selected the run with the highest likelihood. Finally, we compared the cross-validation value from each estimation to determine the K that best fit our data.”

-More details are needed for Local ancestry estimation. The authors need to explicitly say which samples were used in each of the parental populations. Was SHAPEIT2 run in the Population phasing or in the haplotype reference version? Was RFMIX run with one or several EM iterations? What threshold was used in the forward backward values? Because masking is done by making sites missing, is there a missing threshold applied? Again, it needs to be specified how the ADMIXTURE to corroborate that the masking scheme worked was run.

We added a more explicit text for Local Ancestry estimation in line 548: “The masked dataset was generated as follows. First, we generated a reference population panel composed of 50 Native Americans from the MAIS cohort without evidence of recent admixture with continental groups (Africans and European populations) identified in the admixture $K=4$ analyses, 50 Europeans, and 50 Africans derived from the 1000 Genomes Project Phase 3^(ref.62). Except for the Reich data, which were previously masked⁸, the reference dataset was merged individually with each dataset of Mexican Indigenous populations, yielding an SNV intersect of 548,310 for the MAIS cohort genotyped with Affymetrix Genome-Wide Human SNP Array 6.0; 214,968 for the MAIS cohort genotyped with the Illumina HumanOmni 2.5-4v1_B SNP array; 505,024 for the dataset reported by Moreno-Estrada et al², and 303,609 for the dataset reported by Silva-Zolezzi et al¹². Prior to performing local ancestry, each set was phased using SHAPEIT V 2.17^(ref.63) with default parameters. The local ancestry estimation was performed using RFMix¹⁵ with two EM iterations and a forward-backward threshold of 0.9. Non-homozygous native tracts identified by local ancestry estimation were then masked in each individual by setting the genotypes to missing, admitting a maximum threshold of 40% of the genome masked for each sample. After masking, we merged all datasets, including the masked set from Reich et al⁸, yielding a total of 3,490 individuals from 218 populations and 61,393 autosomal SNVs. To test the accuracy of masking, we ran admixture analysis in this dataset with $K=4$ with the previously specified parameters (Supplementary Fig. 2, lower panel).”

-Effective size calculation: What 42 populations? Selected on the basis of what criteria?

We added this text to be clearer in the dataset and the criteria to select the populations for N_e estimation.

In line 573: “The second masked dataset included the masked datasets from the MAIS cohort and Moreno-Estrada et al², both genotyped using the Affymetrix Genome-Wide Human SNP Array 6.0, yielding a total of 996 individuals from 60 populations and 504,581 autosomal SNVs (Supplementary Table 1).”

In line 624: “The effective population sizes of Indigenous Mexican populations were estimated in a set of 48 selected populations from the second masked dataset (Supplementary Table 3). The populations were chosen if the genotyping platform was Human Affymetrix array 6.0 and if there was a minimum sample size of 10. This ensured enough SNP density and the minimum sample size”

-Maximum likelihood tree. What samples with 99.9% Native American ancestry were selected after masking? If so, was a missingness filter applied? Of how much and how many sites passed it? The analysis would be more meaningful if run with some migrations.

To be more explicit in the process to determine the Maximum likelihood tree, we added the following text:

At line 144: “We detected 325 Indigenous samples from the MAIS cohort with at least 0.99 Native American ancestry (Supplementary Fig. 2, upper panel)”

At line 266: “We then combined the 325 indigenous samples with seven ancient genomes from America (Supplementary Table 6), yielding a total of 111,586 autosomal SNVs.”

At line 663: “A maximum likelihood tree was constructed using TreeMix V1.13^(ref.74) based on the Ancient and Modern Native American datasets.”

Figures:

Figure 1: The caption needs to describe better the figure. I think they mistake “branches” by “brackets”. A legend that connects colors with language family is needed, interpretation is quite difficult without it. There should be more intuitive way to distinguish in the figure which samples are from Meso and which are from Aridoamerica. With the current way of displaying it is not intuitive at all. % Variance should be shown in each PC.

Fig 1 was modified as requested.

Figure 2: Again, a better way to intuitively distinguish at first sight between Meso and Arido populations is needed. For panel b) Can populations be sorted in a explicit criteria? It is not clear what the sorting is meant to represent, especially because it is very difficult to read the very tiny population labels. Panel c) “Numbers between branches” -> What branches?. Also, please show the Y axis and specify the value of K.

Fig 2 was modified as requested. Panel (b) was moved to panel (a) in the revised version and was sorted in a north-south gradient to be easy to follows. In panel (c) we changed the figure and added a superscript number to denote the references.

Figure 3: Panel b) What are colors representing and what are the bars? Longtherm -> Longterm. Panel c) Same comment as panel b in Figure2.

To be clearer in panel (b) we follow the order, colors and shapes from figure 1 and panel (c) was reordered following as panel (a) from Fig 3.

Figure 4: Panel a) State what the ML tree is based on (allele frequencies?), show the residual table. Panel b and c) Same comment as Fig.3 panel b. Can they put above plots in c) and d) the topology tested to illustrate what deviations from $D=0$ mean? They do not seem to discuss in the main text why Pame, Huastecos and others have significant (I assume significance but can't be sure given it is not mentioned what the length of the error bars represent, $3de\ 3.3se?$) valued of $D<0$. For panel

d) It would be more informative to have the somehow represented what populations constitute the AA and which ones the MA.

In panel (a) we added the following statement: "Maximum likelihood tree inferred from allele frequency" and added the residual plot. Panels (b) & (c) we added the tested topology above the figures (now panel (b)), Panels (c) & (d) we sort the populations follows the order, colors and shapes from figure 1, also we defined the errors bars. We added a more robust discussion about these results (see page 22 of this document) and added the following paragraph at line 301:

"This is particularly the case for populations from Aridoamerica, with the exception of Cucapa and Seri, as well as some Mesoamerican populations, such as Nahuatl from Puebla, Otomi from Hidalgo, Mixteco Costa, Chocholteco, and Mixe. Although we only found significant results ($|Z| \geq 3.2$) when Tzotzil was used in the comparison, we observed a similar trend when Aymara was used in the test (Fig. 4b and Supplementary Fig. 16c and d). These results suggest that some of the Indigenous populations in our dataset carry ancestry from a population that split before the Anzick-1 individual. Previous studies have suggested that the Mixe carry additional ancestry from an unknown population related to the SNA/ANC-A branch that split above the Anzick-1 individual⁴⁷. Our results are consistent with this observation, suggesting that other populations from Aridoamerica and Mesoamerica may carry this ancestry (Fig. 4b and Supplementary Fig. 16c and d)."

María C. Ávila Arcos

D statistics in the form D(Yoruba, Mesoamerican; Athabaskan, Aridoamerican)

D statistics in the form D(Yoruba, Mesoamerican; Lucier, Aridoamerican)

Ne estimated by IBD tracks in populations without masking step

Ne estimated by IBD tracks in populations with 99 % Native American ancestry

D-statistic in the form of D(Yoruba, Anzick-1; Mexican Native, Karitiana from Reich⁸)

Reviewers' Comments:

Reviewer #1:

Remarks to the Author:

I have reviewed the revised manuscript and the authors' responses. I consider the article has substantially improved and I particularly appreciate the changes to the structure, which make it easier for the reader to grasp the main takeaways. I have just one additional comment and a few minor remarks.

I noticed the data from this study will be made available upon request. I would like to suggest the authors to deposit their data in a public repository which allows for controlled access, such as the EGA. In this case, the authors could submit an anonymised dataset, together with a sharing policy specifying the particular research topics for which the data will be made available (depending on the consent provided by each study participant).

l125. Please spell out the full name of the "MAIS" cohort (first mention in the text).

l191. Stating that a given value of K best fits the data could be misleading. I suggest to only state that $k=9$ yields the lowest cross-validation error.

l242. I understand these estimates assume a clean split between populations. Please specify whether or not this is the case.

l261. Please replace 325 genomes with 325 individuals (these are SNP array data)

l580. fastq files

Reviewer #2:

Remarks to the Author:

In the revised version of the paper "Landscape of genomic diversity in Mexican Indigenous populations: insights into the peopling of the Americas" the authors consolidated the methods section and streamlined the discussion and description of the main results. The inclusion of analysis with ancient genomes helps to ground the discussion point about the peopling of the Americas. The authors greatly improved the paper which now proposes clearer and relevant analysis.

I still have a few comments that I hope the authors can take into account.

From the map, Huichol and Cora are found in the Mesoamerican section, but in the table they are listed as Aridoamerican. Same is true for Mexicanero, if I interpret correctly that the cross of Mexicanero is overlapping the symbol of Huichol. I cannot find the new Tepehuano sample on the map.

Linguistic structure and Fst NJ tree.

The linguistic clustering that I see includes one strongly defined linguistic cluster for Maya speakers (except Huasteco, as noted by the authors when mentioning the similarity of groups in the huasteca region). With the proximity of Zoque to the Mayan cluster, and looking at the geographic distribution of the samples, this grouping seems to be the result again of geographic proximity, rather than a particular linguistic connection. As the other groups do not correspond to particular linguistic clustering, I would not stress the linguistic affiliation influencing genetic proximity as a result of the paper. The northern cluster in fact includes the aridoamerican populations: Seri, cochimi-Yumana speakers (except Kuahl), and part of the Uto-Aztecan with speakers of the Tepiman, Cahitan and Guajiro-Tarajumare subfamilies. Except Kuahl, these populations are the ones in the aridoamerican group, so this is a geographic cluster and not a linguistic cluster. Interestingly, some Uto Aztecan from Mesoamerica form a separate branch (the one at the top of the tree): these are other speakers of the already mentioned Tepiman (against a close linguistic clustering), together with speakers of the Corachol and Aztecan family. Is there something unique of these groups, that can justify their

separation from the other populations? Finally, other speakers of the Aztecan family (Nahuatl) are included in other branches scattered with other mesoamerican groups.

The third proposed cluster includes most of the Otomanguean speakers (suggesting indeed genetic cohesiveness for this particular language family), but it should be mentioned that there are genetically close also to the Mixe and one Nahuatl. (note: I understand that the authors are using the classification from INALI, and I appreciate their clarification, but as I am not familiar with that I refer to Glottolog as a reference for nomenclature and classification).

Summarizing, I do not see effect of preferential genetic connections mediated by similar linguistic affiliation: the correspondence described can be explained by geography alone. I would mention this also in the discussion, stating that this effect prevents to make strong assumptions of preferential cultural connections and gene-flow between speakers of the same language family or linguistically related groups. The authors already toned down previous assumptions of language structure, but I believe that this part in the discussion needs further consideration.

Geographic barriers

I appreciate the authors' work in giving more explanations on sources of geographic structuring, but a few clarifications are still needed and would help to ground the discussion. Explain what is and where it is the Eje Neovolcanico. Geographic barriers are described and localized when the authors talk about the semidesertic area of the north and the jungle near Lacandon. Please give further examples that can link the other "mountain ranges, canyons and jungle regions" as mentioned in line 350. For example, it is not clear if we must think about existing barriers separating populations (in this case, where are the major barriers? North south mountain ranges? In this case should be expect two blocks of structure, like a west vs. east gene pool?) or if we must refer to a scattered and diverse ecogeographic map where populations living in the jungle are isolated from populations living in the mountains or in the semidesertic areas (in this case we should expect different patches of genetic diversity). These 2 hypotheses on geographic barriers/geographic structure would be different from a simple Isolation By Distance model, where there is a gradient of genetic diversity shaped by geography (no geographic barriers needed for this): a similar scenario to the one described in lines 355-360.

Line 379: are there different demographic histories in each population, or all populations are equally affected by the post European contact reduction in population size? Or are these different demographic histories occurring before the European contact?

Lines 425-435 are a new important point for the paper. Could these two hypotheses be summarized in a Figure, with different population split schemes and approximate time scale?

Finally, I wonder if some sentences can be added to discuss the presence of drifted population that show their differentiation in PCA and Figure S5, in comparison to their level of endogamy and their N_e . some populations seems drifted, with long F_{ST} , long ROH and small N_e , while other populations correspond to separate clusters in Figure S5, but have shorter ROH and larger N_e (e.g. Tarahuamara). is there something to add to the discussion to characterize populations that might be genetically differentiated due to a history of separate cultural trajectories, complex societies, expanding empires vs. populations that experienced isolation and drift?

Tree mix labels are difficult to associate to branches. Can the tree be zoomed in slightly?

Figure S4 (ADMIXTURE) : for better readability, would it be possible to make the admixture plot bigger, and the CV plot smaller?

Reviewer #4:
Remarks to the Author:

In "Landscape of genomic diversity in Mexican Indigenous populations: insights into the peopling of the Americas" Humberto Garcia Hortiz, Francisco Barajas-Olmos, Cecilia Conteras-Cubas and colleagues generate genome-wide data for 716 individuals belonging to 60 ethnic groups from Mexico. Combining the new data with those to Mexican and worldwide populations, and harnessing mostly genotype based methods, (rather than haplotype-based methods, with the exception of IBDNe analysis), they found that the genetic structure is correlated to geographic location of individuals, a result that emerged already in Moreno-Estrada et al. 2014, and therefore not novel. However, for most of the analyses, the authors use a low number of markers, which could potentially affect the level of analysis of details.

Although the data is valuable, and some of the results interesting I honestly don't think they have the sufficient level of originality at this stage and a certain number of additions and clarifications should be included.

The authors claim that the data related to the manuscript are "The data that support the findings of this study are available upon request to the corresponding authors: L.O. or H.G.-O. The data are not publicly available because they contain information that could compromise research participant privacy/consent". This, alone, is, in my opinion sufficient to decline the publication of the manuscript in such an influential journal. I firmly believe that data sharing is essential for two perspectives: reproducibility and fairness. The authors use almost 3,000 publicly available individual genotype data, without which all the analysis would be impossible to perform. Are these not containing "information that could compromise research participant privacy/consent"? I strongly suggest the authors provide the data available for research purposes, or at least to deposit them on one of the many archives which allow for data requests (EGA,dbgap,...).

Estimates of divergence times among populations exploiting NE estimates based on LD-patterns and F_{ST} between populations suggests that most of the differentiation in Mexico occurred between 10k and 4k years ago. However, such methods assume clean split models which are very unlikely to have occurred in Human evolution. Although I find these results interesting, a discussion on possible limitations of the methods, and how these results fit with archaeological and ancient DNA data is lacking.

Furthermore, the large bulk of results rely on analysis performed on allele frequency and an assessment of IBD among populations or other haplotype-based methods such as Chromopainter/Finestructure/Globetrotter or Mosaic would have been crucial to better understand the genetic structure of Mexican populations at a fine scale.

Minor remarks:

Line 99. Why from North to South and not vice versa?

L 107. I honestly don't think there exist areas most important to study, implying there are some least important...

L. 118. Although technically correct, the world whole genome level might be misleading, I needed to check the Materials and Methods section to get if the data were coming from genotyping arrays or sequencing. I would suggest the use of genome-wide data.

L. 150. From Supplementary figure 2, it looks like, despite the masking methods, that some individuals still have Eurasian ancestry, which could in principle affect the downstream analysis. I suggest the authors verify the effective performance of the masking using the f_4 , and eventually remove individuals which are still showing higher affinity to non american populations.

L.172 Higher compared to which populations?

L. 180 Despite being a very interesting result, the authors do not discuss, what are the implications of this finding. Are these due to cultural switch, migration admixture? How the observed pattern might

be explained?

L. 298 Figure 4a is a Treemix analysis. Is this what the authors are referring to?

Dear reviewers,

We would like to thank you for taking the time to review our manuscript one more time. All your comments and suggestions were very helpful to improve the quality of our investigation.

The main improvement of our work was the inclusion of IBD network analyses which helped us to identify spatiotemporal connections between indigenous populations from different regions of Mexico and reinforce the previously identified gene flow with Treemix analyses.

Please find below (in blue) more detailed responses to each of your original comments. We hope that this revised version will now be suitable for publication in Nature Communications.

Reviewer #1 (Remarks to the Author):

I have reviewed the revised manuscript and the authors' responses. I consider the article has substantially improved and I particularly appreciate the changes to the structure, which make it easier for the reader to grasp the main takeaways. I have just one additional comment and a few minor remarks.

I noticed the data from this study will be made available upon request. I would like to suggest the authors to deposit their data in a public repository which allows for controlled access, such as the EGA. In this case, the authors could submit an anonymised dataset, together with a sharing policy specifying the particular research topics for which the data will be made available (depending on the consent provided by each study participant).

Thanks a lot for your comments and suggestions.

Since indigenous people are considered as vulnerable population, the recruitment of individuals was completed with prior approval of indigenous leader of each community and with the support of the National Commission for the Development of Indigenous Communities of Mexico (CDI, from the Spanish "Comisión Nacional para el Desarrollo de Pueblos Indígenas"). During the presentation of the project, we agreed with the communities to conduct ourselves with the values of equity, respect, and mutual collaboration. Highlighting that all their samples and data generated are protected from any improper use. Therefore, we modified the statement, as follows:

"In alignment with the Institutional Review Board approval and the individual informed consents forms, as well as to avoid compromising the participant's privacy, the whole genotype data from the new Indigenous individuals presented here, are available through a data-access agreement."

125. Please spell out the full name of the "MAIS" cohort (first mention in the text).

MAIS cohort was spelled out when it appears for the first time in the text (line 123).

1191. Stating that a given value of K best fits the data could be misleading. I suggest to only state that k=9 yields the lowest cross-validation error.

The sentence was changed as you suggested. (line 188).

1242. I understand these estimates assume a clean split between populations. Please specify whether or not this is the case.

We added the following phrase at line 238: "...assuming a clean split between them."

1261. Please replace 325 genomes with 325 individuals (these are SNP array data)

We exchange genomes by individuals, as suggested. (line 284)

1580. fastq files.

The text was corrected. (line 628)

Reviewer #2 (Remarks to the Author):

In the revised version of the paper "Landscape of genomic diversity in Mexican Indigenous populations: insights into the peopling of the Americas" the authors consolidated the methods section and streamlined the discussion and description of the main results. The inclusion of analysis with ancient genomes helps to ground the discussion point about the peopling of the Americas. The authors greatly improved the paper which now proposes clearer and relevant analysis.

Thanks a lot for your comments

I still have a few comments that I hope the authors can take into account.

From the map, Huichol and Cora are found in the Mesoamerican section, but in the table they are listed as Aridoamerican. Same is true for Mexicanero, if I interpret correctly that the cross of Mexicanero is overlapping the symbol of Huichol. I cannot find the new Tepehuano sample on the map.

Thank you for your observation, you are right.

We have properly classified the Huichol and Cora in the supplementary table 1 and in the rest of the text. We also updated the map to better show each population.

Linguistic structure and Fst NJ tree.

The linguistic clustering that I see includes one strongly defined linguistic cluster for Maya speakers (except Huasteco, as noted by the authors when mentioning the similarity of groups in the huasteca region). With the proximity of Zoque to the Mayan cluster, and looking at the geographic distribution of the samples, this grouping seems to be the result again of geographic proximity, rather than a particular linguistic connection. As the other groups do not correspond to particular linguistic clustering, I would not stress the linguistic affiliation influencing genetic proximity as a result of the paper. The northern cluster in fact includes the aridoamerican populations: Seri, cochimi-Yumana speakers (except Kuahl), and part of the Uto-Aztecan with speakers of the Tepiman, Cahitan and Guajiro- Tarajuamare subfamilies. Except Kuahl, these populations are the ones in the aridoamerican group, so this is a geographic cluster and not a linguistic cluster.

Interestingly, some Uto Aztecan from Mesoamerica form a separate branch (the one at the top of the tree): these are other speakers of the already mentioned Tepiman (against a close linguistic clustering), together with speakers of the Corachol and Aztecan family. Is there something unique of these groups, that can justify their separation from the other populations? Finally, other speakers of the Aztecan family (Nahuatl) are included in other branches scattered with other mesoamerican groups.

The third proposed cluster includes most of the Otomanguean speakers (suggesting indeed genetic cohesiveness for this particular language family), but it should be mentioned that there are genetically close also to the Mixe and one Nahuatl. (note: I understand that the authors are using the classification from INALI, and I appreciate their clarification, but as I am not familiar with that I refer to Glottolog as a reference for nomenclature and classification).

Summarizing, I do not see effect of preferential genetic connections mediated by similar linguistic affiliation: the correspondence described can be explained by geography alone. I would mention this also in the discussion, stating that this effect prevents to make strong assumptions of preferential cultural connections and gene-flow between speakers of the same language family or linguistically related groups. The authors already toned down previous assumptions of language structure, but I believe that this part in the discussion needs further consideration.

You are right. To avoid confusion about the relationship among populations and genetic distances, we made the next changes in the text:

In Results section the next paragraphs were removed:

“A midpoint rooted neighbor-joining (NJ) tree based on the pairwise-FST population distances showed a correspondence between geographic distribution and linguistic classification proposed by the National Institute of Indigenous Languages in Mexico (INALI; from the Spanish Instituto Nacional de las Lenguas Indígenas)”

“The NJ tree topology revealed three major clades, with high clustering of the main linguistic families: Yuto-Nahua and Cochimi-Yumana in the north, Oto-Mange in the center/south, and Mayan in the southeast”

In Discussion section the next paragraph was removed:

“Otherwise, the pattern observed in the NJ tree, in which individuals from related languages cluster together (Fig. 2a), is compatible with previous anthropological studies showing that human cultural practices, such as language and human cooperation, as well as many other cultural features, have modified environmental conditions, triggering changes in allele frequencies and contributing to differentiation of the populations”

Geographic barriers

I appreciate the authors' work in giving more explanations on sources of geographic structuring, but a few clarifications are still needed and would help to ground the discussion. Explain what is and where it is the Eje Neovolcanico. Geographic barriers are described and localized when the authors talk about the semidesertic area of the north and the jungle near Lacandon. Please give further examples that can link the other “mountain ranges, canyons and jungle regions” as mentioned in line 350. For example, it is not clear if we must think about existing barriers separating populations (in this case, where are the major barriers? North south mountain ranges? In this case should be expect two blocks of structure, like a west vs. east gene pool?) or if we must refer to a scattered and diverse ecogeographic map where populations living in the jungle are isolated from populations living in the mountains or in the semidesertic areas (in this case we should expect different patches of genetic diversity). These 2 hypotheses on geographic barriers/geographic structure would be different from a simple Isolation By Distance model, where there is a gradient of genetic diversity shaped by geography (no geographic barriers needed for this): a similar scenario to the one described in lines 355-360.

We modified the text to be clearer about the “Eje Neovolcanico” as follows:

“...located around the Neovolcanic axis at the central part of Mexico.”

Starting at line 373 we modified the text about the geographic barriers hypothesis as follows:

“Altogether, these results support previous hypotheses^{2,34,35} suggesting that the geographic barriers observed in the Mexican territory have played a major role in shaping the observed patterns of genetic structure in present-day Indigenous populations”

Line 379: are there different demographic histories in each population, or all populations are equally affected by the post European contact reduction in population size? Or are these different demographic histories occurring before the European contact?

In order to be clearer, we included the observation about the bottleneck before European contact (lines 397-400) as follows:

“Overall, our findings show, for the first time, that the strength and timing of the contraction observed in Indigenous populations was not localized to a particular population but, instead, has been widespread in all tested populations, and in some cases, they took place before the European contact”

Lines 425-435 are a new important point for the paper. Could these two hypotheses be summarized in a Figure, with different population split schemes and approximate time scale?

Finally, I wonder if some sentences can be added to discuss the presence of drifted population that show their differentiation in PCA and Figure S5, in comparison to their level of endogamy and their N_e . some populations seems drifted, with long F_{ST} , long ROH and small N_e , while other populations correspond to separate clusters in Figure S5, but have shorter ROH and larger N_e (e.g. Tarahuamara). is there something to add to the discussion to characterize populations that might be genetically differentiated due to a history of separate cultural trajectories, complex societies, expanding empires vs. populations that experienced isolation and drift?

We included an IBD analysis, which revealed several connections between populations from different regions and helped us to link some population movements and relationships.

The text added about this start at line 423:

“IBD networks could reflect spatiotemporal dynamics between populations from different regions. Network visualization by intermediate track sizes suggests that such interregional movements occurred around 500-1,500 years ago or more as revealed between Tarahumara and Guarijio corresponding to the North of Mexico and Mayan groups from the Southeast region or with Matlaltzinca or Triqui from the South region (Fig. 4 and Supplementary Fig. 11a). Historically, these findings could be supported by the hypothesis of the agriculture introduction to the North region around 2000 years ago, due to a movement of Indigenous populations from the Center and South of Mexico^{50,51}. Meanwhile, recent connections around 0-500 years ago between indigenous populations from different regions were also suggested by IBD network, through the large track size analysis (Supplementary Fig. 11b). Dynamic movements have been largely observed

between indigenous populations in recent times. An example of this occurred during “The Porfiriato”, a dictatorial era in Mexico during 1876-1911AD, when some ethnic groups from the North were forced to work in the Southeast of the country, especially in the Mayan region⁵². Further studies are still needed to clarify the contribution of cultural traditions and transitions may have had on the genetic structure of present-day Indigenous peoples inhabiting the Mexican territory.

Tree mix labels are difficult to associate to branches. Can the tree be zoomed in slightly?

Thank you for your observation. We improved the Treemix supplementary figure by zoomed the tree and labels.

Figure S4 (ADMIXTURE): for better readability, would it be possible to make the admixture plot bigger, and the CV plot smaller?

We decided to divide the figure to properly show both plots, now the admixture plot is Figure S4 and the CV plot is Figure S5.

Reviewer #4 (Remarks to the Author):

In “Landscape of genomic diversity in Mexican Indigenous populations: insights into the peopling of the Americas” Humberto Garcia Hortiz, Francisco Barajas-Olmos, Cecilia Conteras-Cubas and colleagues generate genome-wide data for 716 individuals belonging to 60 ethnic groups from Mexico. Combining the new data with those to Mexican and worldwide populations, and harnessing mostly genotype based methods, (rather than haplotype-based methods, with the exception of IBDNe analysis), they found that the genetic structure is correlated to geographic location of individuals, a result that emerged already in Moreno-Estrada et al. 2014, and therefore not novel. However, for most of the analyses, the authors use a low number of markers, which could potentially affect the level of analysis of details.

Although the data is valuable, and some of the results interesting I honestly don’t think they have the sufficient level of originality at this stage and a certain number of additions and clarifications should be included.

Thanks for your comments, we responded to each one of theme in more detail.

The authors claim that the data related to the manuscript are “The data that support the findings of this study are available upon request to the corresponding authors: L.O. or H.G.-O. The data are not publicly available because they contain information that could compromise research participant privacy/consent”. This, alone, is, in my opinion sufficient to decline the publication of the manuscript in such an influential journal. I firmly believe

that data sharing is essential for two perspectives: reproducibility and fairness. The authors use almost 3,000 publicly available individual genotype data, without which all the analysis would be impossible to perform. Are these not containing “information that could compromise research participant privacy/consent”? I strongly suggest the authors provide the data available for research purposes, or at least to deposit them on one of the many archives which allow for data requests (EGA,dbgap,...).

It is not possible to upload the genomic data of these vulnerable populations in public repositories, due that when this project started this specific point was a request from indigenous leaders to allow us to go ahead with our investigation as stated in the consent form. As you probably know, this is in accordance with the guide published by Claw K.G., et al. (2018; 9:2957 | DOI: 10.1038/s41467-018-05188-3). Nevertheless, this does not mean that the data from new genotyped individuals will not be available for further investigations and to clarify this we modified the data availability statement as follows:

“In alignment with the Institutional Review Board approval and the individual informed consents forms, as well as to avoid compromising the participant’s privacy, the whole genotype data from the new Indigenous individuals presented here, are available through a data-access agreement.”

Estimates of divergence times among populations exploiting NE estimates based on LD-patterns and FST between populations suggests that most of the differentiation in Mexico occurred between 10k and 4k years ago. However, such methods assume clean split models which are very unlikely to have occurred in Human evolution. Although I find these results interesting, a discussion on possible limitations of the methods, and how these results fit with archaeological and ancient DNA data is lacking.

We included several sentences of the limitation of this approach:

Discussion Section, Line 414. We included the following text: “Nevertheless, the assumption of a clean split between populations from Aridoamerica and Mesoamerica could underestimate the observed T due to the relatively recent gene flow between populations from both cultural areas.”

Discussion Section, Line 479. “A limitation of the present study is that genetic data were obtained using a commercial microarray that is not designed for Native American populations, which could hinder the resolution of the inferences”

Regarding on the comparison with ancient DNA, in our knowledge there are not much data of ancient DNA from Mexican regions to allow us to compare them with our data. The discussion of the possible occurrence of these Divergence times with cultural shifts in both regions could be found in the Discussion section, i.e., in line 401: “On the other hand, the estimated T of 3.96 to 9.47 ka ago and 4.84 to 10.15 ka ago between populations from

Aridoamerica and Mesoamerica (Fig. 3c and 3d) overlaps with the beginning and establishment of agriculture^{33,36,43-46}

Furthermore, the large bulk of results rely on analysis performed on allele frequency and an assessment of IBD among populations or other haplotype-based methods such as Chromopainter/Finestructure/Globetrotter or Mosaic would have been crucial to better understand the genetic structure of Mexican populations at a fine scale.

**Thank for your suggestion, it improved the quality of our manuscript.
We included an IBD analysis, whose results are shown in Figure 4, Supplementary Fig 11 and supplementary tables 7-11**

Minor remarks:

Line 99. Why from North to South and not vice versa?

You are right, we added “...and vice versa”

L 107. I honestly don't think there exist areas most important to study, implying there are some least important...

We removed this sentence from the revised version of the manuscript (Line 107)

L. 118. Although technically correct, the world whole genome level might be misleading, I needed to check the Materials and Methods section to get if the data were coming from genotyping arrays or sequencing. I would suggest the use of genome-wide data.

We modified the text as follows (Line 115): “at the genome-wide level”

L. 150. From Supplementary figure 2, it looks like, despite the masking methods, that some individuals still have Eurasian ancestry, which could in principle affect the downstream analysis. I suggest the authors verify the effective performance of the masking using the f_4 , and eventually remove individuals which are still showing higher affinity to non american populations.

Thanks for your suggestion, nevertheless as you can note there is no change in the inferences performed with the masked data set and the analysis based on 99 % native American ancestry individuals. These can be observed in the PCA analyses (Fig.1b, and supplementary figure 13). This is also observed in the determination of N_e from IBD tracks in which the estimation in masked (supplementary figure 8), and 99% ancestry data sets, yields the same results (figure attached at the end of this document).

L.172 Higher compared to which populations?

It is not referred to comparisons among populations, it is based on the statistic analyses, which are indicating that phenomena.

L. 180 Despite being a very interesting result, the authors do not discuss, what are the implications of this finding. Are these due to cultural switch, migration admixture? How the observed pattern might be explained?

According with the recommendations of reviewer #2, this asseveration was removed from the main manuscript.

L. 298 Figure 4a is a Treemix analysis. Is this what the authors are referring to?

Thanks for your observation, it was a mistake. We corrected the mistake and the proper figure was referred as (Fig. 5b).

Ne estimated by IBD tracks in populations with 99 % Native American ancestry

Reviewers' Comments:

Reviewer #2:

Remarks to the Author:

In these revisions, the authors replied point by point to all the issues raised by the reviewers and made changes accordingly in the manuscript. The manuscript now is consistent and solid.

I agree with the authors about the importance of not sharing the data from ethnic minorities and indigenous groups in a public database. At the present time, while it is not a perfect solution, is the best practice we can adopt to protect the privacy and the potential discriminatory use of the data, and to sustain access and representation in genetic studies. The same protocol is widely applied by authors who work with genetic data from such sensitive nature.

A very minor comment about the new IBD section.

I appreciate the inclusion of IBD analysis, which adds information to the paper.

The use of intermediate and large bins for IBD fragments is appropriate. Nevertheless, the reference to an exact time frame for the fragment size is not supported by specific case studies in the Americas. The authors cite Ralph and Coop, which are one of the main references for this kind of analysis and statistical reasoning, but who applied it only to Europeans. They suggest that "blocks longer than 4 cM come from 500–1,500 years ago, and blocks longer than 10 cM from the last 500 years.", while your threshold cut is at 5cM. Similar analysis was implemented by Harris et al. PNAS 2018 with a Peruvian dataset, but the connection between size of the fragment and generations since common ancestor was never tested in the Americas. I argue that the same IBD size fragments should be weighted differently in Europe or Africa and in the Americas, with the Americas having a deeper divergent times for the same size, an effect resulting from the strong initial bottleneck at the colonization of the continent. According to this reasoning, the proposed time bin in the Americas would be stretched towards deeper times, which would in turn allow for compatibility with the events of 2000 years ago that you outline (line 431) - therefore dropping the time limit of 500-1,500 years would be indeed fitting. The other context you proposed is about events from the 1800, which is of course possible for bins of >10cM (which include also potentially very large fragments from a few generations ago).

For the moment, without specific testing of the fragment sizes and contact events within the Americas available, I would avoid direct association between IBD bins and precise time frame in generations ago. I would keep mention of indicative time limits by referring to studies applied to non-American populations.

in line 712 "As described in Ioannidis G et al" .. reference missing.

Reviewer #4:

Remarks to the Author:

The authors have responded to all the points raised in my previous round of revision and therefore the manuscript is now suitable for publication in Nature Communications.

Dear reviewers,

We would like to thank you for taking the time to review our manuscript one more time. We also thank you for all your previous and present comments that improved our manuscript's quality and for your positive opinion about our work.

Here we answer in detail your last comments:

Reviewer #2 (Remarks to the Author):

In these revisions, the authors replied point by point to all the issues raised by the reviewers and made changes accordingly in the manuscript. The manuscript now is consistent and solid.

I agree with the authors about the importance of not sharing the data from ethnic minorities and indigenous groups in a public database. At the present time, while it is not a perfect solution, is the best practice we can adopt to protect the privacy and the potential discriminatory use of the data, and to sustain access and representation in genetic studies. The same protocol is widely applied by authors who work with genetic data from such sensitive nature.

Thank you for all your comments and for finding our manuscript consistent and solid.

A very minor comment about the new IBD section.

I appreciate the inclusion of IBD analysis, which adds information to the paper. The use of intermediate and large bins for IBD fragments is appropriate. Nevertheless, the reference to an exact time frame for the fragment size is not supported by specific case studies in the Americas. The authors cite Ralph and Coop, which are one of the main references for this kind of analysis and statistical reasoning, but who applied it only to Europeans. They suggest that "blocks longer than 4 cM come from 500–1,500 years ago, and blocks longer than 10 cM from the last 500 years.", while your threshold cut is at 5cM. Similar analysis was implemented by Harris et al. PNAS 2018 with a Peruvian dataset, but the connection between size of the fragment and generations since common ancestor was never tested in the Americas. I argue that the same IBD size fragments should be weighted differently in Europe or Africa and in the Americas, with the Americas having a deeper divergent times for the same size, an effect resulting from the strong initial bottleneck at the colonization of the continent. According to this reasoning, the proposed time bin in the Americas would be stretched towards deeper times, which would in turn allow for compatibility with the events of 2000 years ago that you outline (line 431) - therefore dropping the time limit of 500-1,500 years would be indeed fitting. The other context you proposed is about events from the 1800, which is of course possible for bins of >10cM (which include also potentially very large fragments from a few generations ago). For the moment, without specific testing of the fragment sizes and contact events within the Americas available, I would avoid direct association between IBD bins

and precise time frame in generations ago. I would keep mention of indicative time limits by referring to studies applied to non-American populations.

We agree with you, we made changes in the manuscript in order to be cautious about these affirmations, avoiding to make direct association of the IBD bins with specific events.

Also, we added a limitation of this estimation in the discussion section as follows:

“Nevertheless, we should be cautious in our interpretation of historic events associated with these IBD patterns, because the time frames were inferred in European populations and could be different in other populations such as Native Americans²⁷”.

in line 712 “As described in Ioannidis G et al” .. reference missing.

Thank you for notice our omission, we added the paper by Ioannidis G et al to our reference list.

Reviewer #4 (Remarks to the Author):

The authors have responded to all the points raised in my previous round of revision and therefore the manuscript is now suitable for publication in Nature Communications.

Thank you for your comments and for finding this revised version of our work suitable for publication in Nature Communications.